

# Implementing a coral reef CaCO₃ production module in the iLOVECLIM climate model

Nathaelle Bouttes[1], Lester Kwiatkowski[2], Manon Berger[1,3], Victor Brovkin[4], Guy Munhoven[5]

[1]Laboratoire des Sciences du Climat et de l'Environnement, LSCE/IPSL, CEA-CNRS-UVSQ, Université Paris-Saclay, Gif-sur-Yvette, France
[2]LOCEAN Laboratory, Sorbonne Université-CNRS-IRD-MNHN, Paris, 75005, France
[3]LMD-IPSL, CNRS, Ecole Normale Supérieure/PSL Res. Univ, Ecole Polytechnique, Sorbonne Université, Paris, 75005, France
[4]Max Planck Institute for Meteorology, Hamburg, Germany; also CEN, University of Hamburg, Germany;
[5]Dépt. d'Astrophysique, de Géophysique et d'Océanographie, Université de Liège, B-4000 Liège, Belgique

*Correspondence to*: Nathaelle Bouttes (Nathaelle.bouttes@lsce.ipsl.fr)

**Abstract.** Coral reef development is intricately linked to both climate and the concentration of atmospheric CO₂, specifically through temperature and carbonate chemistry in the upper ocean. In turn, the calcification of corals modifies the concentration of dissolved inorganic carbon and total alkalinity in the ocean, impacting air-sea gas exchange, atmospheric CO₂ concentration, and ultimately the climate. This retroaction between atmospheric conditions and coral biogeochemistry can only be accounted for with a coupled coral-carbon-climate model. Here we present the implementation of a coral reef calcification module into an Earth System model. Simulated coral reef production of the calcium carbonate mineral aragonite depends on photosynthetically active radiation, nutrient concentrations, salinity, temperature and the aragonite saturation state. An ensemble of 210 parameter perturbation simulations was performed to identify carbonate production parameter values that optimise the simulated distribution of coral reefs and associated carbonate production. The tuned model simulates the presence of coral reefs and regional-to-global carbonate production values in good agreement with data-based estimates. The model enables assessment of past and future coral-climate coupling on seasonal to millennial timescales, highlighting how climatic trends and variability may affect reef development and the resulting climate-carbon feedback.

## 1 Introduction

Tropical coral reefs are well known for the provisional and cultural ecosystem services they provide, supporting large fisheries (Hoegh-Guldberg, 1999) and a multi-billion-dollar tourism industry (Spalding et al., 2017). However, they also play an





important role in the carbon cycle and hence climate regulation. The production of calcium carbonate by coral reefs consumes total alkalinity and dissolved inorganic carbon in a ratio of 2:1, decreasing pH, increasing $[CO_2]$ and in an open system resulting in outgassing of $CO_2$ to the atmosphere (Gattuso et al., 1999; Bates et al., 2001; Wolf-Gladrow et al., 2007; Suzuki and Kawahata, 2003). On the contrary, dissolution of calcium carbonate has the opposite effect, acting to lower the concentration of atmospheric $CO_2$.

Due to this effect, coral reefs have been proposed as a possible cause of the deglacial $CO_2$ increase from the cold Last Glacial Maximum around 21 000 years ago to the warmer Holocene around 9 000 years ago. During this deglaciation period, the sea level rose by around 120 m (Gowan et al., 2021). It has been hypothesised that this led to the colonization of flooded continental shelf by coral reef, with enhanced global calcification, increasing atmospheric $CO_2$ and acting as a positive feedback on deglacial warming. This hypothesis was first proposed by Berger (1982) and subsequently tested and discussed in several
studies (Opdyke and Walker, 1992; Walker and Opdyke, 1995; Munhoven and Francois, 1996; Kleypas, 1997; Ridgwell et al., 2003; Vecsei and Berger, 2004). Although the scale of coral contribution to the deglacial $CO_2$ rise is not well constrained, its potentially substantial role on interglacial $CO_2$ changes such as those during the Holocene has been demonstrated (Ridgwell et al., 2003; Kleinen et al., 2016, Menviel and Joos, 2012, Brovkin et al., 2016).

As climate is projected to change in the future, so is the extent and distribution of coral reef cover, which is influenced by sea
level, ocean temperature, nutrient concentrations and carbonate chemistry. Studies of long term (> 1,000 years) evolution of the future carbon cycle have mostly focused on the effect of deep-sea sediments (Archer, 2005; Archer et al., 2009), overlooking the potential influence of coral reef changes on tropical shelves.

To understand and evaluate the role of coral reefs in the carbon cycle and their resulting effect on climate, it is necessary to use a carbon-climate model that includes a coral reef carbonate production module. Based on studies investigating the effect
of warming and/or ocean acidification on corals (either in situ or in laboratories), empirical models have been developed to evaluate coral reef changes, regionally or globally (Kleypas et al., 1999; Donner et al., 2005; Buddemeier et al., 2008; Silverman et al., 2009; Pandolfi et al., 2011; Frieler et al., 2012; Kwiatkowski et al., 2015; van Hooidonk et al., 2016). However, most of these focus on the development and bleaching of corals and not explicitly on carbonate production. In addition, they do not take into account the feedback on the rest of the carbon cycle, which would alter the response. Less than
a handful of models of coral reef carbonate production have been developed, and most have shown poor performance compared to observations (Jones et al., 2015). In addition, with the exception of the CLIMBER-2 intermediate complexity model (Kleinen et al., 2016), no coral reef carbonate production model has been coupled to a climate-carbon model. Instead, simulations with climate models have been limited to prescribing DIC and alkalinity fluxes associated with net calcification/dissolution (Ridgwell et al., 2003; Kleinen et al., 2010; Brovkin et al., 2019; O'Neill et al., 2021). Moreover,
using climate model outputs to force coral niche or impact models offline, as has been historically the case, has limitations. Simulated variables from climate models are not always archived at the needed temporal resolution. While annual and monthly outputs are usually available, daily and diel values are often not kept for simulations of more than a century, due to the



associated storage requirement. This prevents precise computation of simulated bleaching events using degree heating weeks and/or accounting for sub-monthly carbonate chemistry variability (Torres et al., 2021; Kwiatkowski et al., 2022). Directly
coupling a coral reef module to a climate model negates such limitations.

Here we have implemented a coral calcification module into the iLOVECLIM carbon-cycle-climate model. iLOVECLIM is an intermediate complexity model well suited for multi-millennial climate simulations, that has already been used in numerous studies addressing changes during the Last Glacial Maximum (Lhardy et al., 2021), past interglacials (Bouttes et al., 2018) or the last 2000 years (Bakker et al., 2022). The coral module described here is based on the ReefHab model (Kleypas, 1995,
1997), but includes several extensions to improve its performance and account for wider process complexity. Specifically, given that warming and heat waves leading to bleaching can severely impact coral reefs (Sully et al., 2019), and ocean acidification can hinder calcification (Chan and Connolly, 2013; Albright et al., 2018), we have incorporated temperature and aragonite saturation state dependent parameterizations of coral reef carbonate production, as well as a bleaching component.

## 2 Methods

We have coupled the iLOVECLIM climate model to a tropical coral reef module. We describe the model, the simulations and data used to select the best parameter sets and validate the new coupled model in modern conditions.

### 2.1 Description of the iLOVECLIM model

iLOVECLIM is an intermediate complexity model including atmosphere (ECBILT), ocean (CLIO), sea ice (LIM) and continental vegetation (VECODE) components inherited from the LOVECLIM model (Goosse et al., 2010). It is also coupled to a carbon cycle module (Bouttes et al., 2015). It has a horizontal ocean resolution of 3° with 20 vertical levels (including 6 levels in the upper 100 m), while the atmosphere is a T21 quasi-geostrophic model with 3 vertical levels. iLOVECLIM is well suited to long duration and large ensemble simulations as it can simulate around 700 years/day on a 7 core CPU.

The ocean carbon cycle, described in Bouttes et al. (2015) is based on a Nutrient-Phytoplankton-Zooplankton-Detritus (NPZD) model (Six and Maier-Reimer, 1996; Brovkin et al., 2002). It includes dissolved inorganic carbon (DIC) and alkalinity (ALK). The air-sea gas exchange of $CO_2$ depends on sea ice coverage, wind speed and the air-sea $p CO_2$ gradient. Surface ocean $p CO_2$ is computed from temperature, salinity, DIC and ALK following Millero (1995). The oxygen surface concentration is prescribed to saturation. The model comprises one phytoplankton type, one zooplankton type, nutrients (nitrates and
phosphates), oxygen, two types of dissolved organic carbon (labile and refractory), particulate organic carbon (POC) and calcium carbonate in the form of calcite ($CaCO_3$) that results from implicit pelagic calcification. Photosynthesis takes place in the euphotic zone in the upper 100 meters. All tracers follow the advection-diffusion scheme of the ocean model, with the exception of POC and $CaCO_3$ which sink and are remineralized at depth with a fixed vertical profile.



## 2.2 Description of the iCORAL coral reef module


The coral reef module, called iCORAL (interactive CORAL reef accumulation module) is a module of calcium carbonate (aragonite) production based on the ReefHab model (Kleypas, 1995; Kleypas, 1997) with several modifications and developments that we describe below. It aggregates the carbonate production of warm water coral reef ecosystems composed of corals, calcareous algae and other calcifiers depending on local variables (Figure 1).

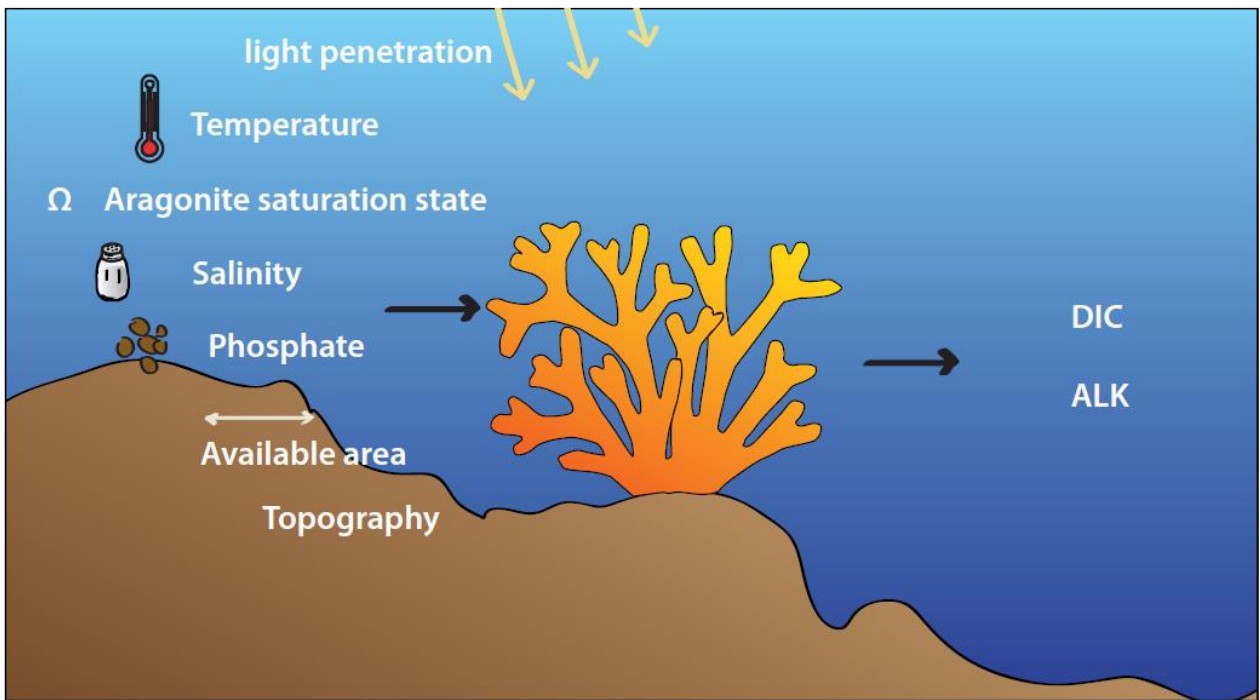


**Figure 1. Summary of all variables governing carbonate production in the iLOVECLIM coral module (left) and the variables that are impacted by carbonate production (right).**

### 2.2.1 Coral habitability

As in ReefHab, iCORAL first computes the coral habitability in each grid cell. Coral carbonate production can take place in a grid cell under the requirement that the following conditions are satisfied:

- The temperature is between 18.1°C and 31.5°C and exceeds 18.1°C throughout the year.
- The salinity is between 30 and 39
- The phosphate concentration is below 0.2 μmol/L



-     The depth $Z$ is shallower than the maximum coral production depth ($Z_{max}$) which depends on attenuation of light in the water column:

$$Z_{max} = \frac{\log\left(\frac{I_{min}}{PAR}\right)}{K_{490}}$$

where $I_{min}$ is a fixed parameter (the minimum light intensity necessary for reef growth) that is optimized during model tuning (Table 1), PAR is the photosynthetically active radiation at the surface (computed by the iLOVECLIM climate model) and

$K_{490}$ is the diffuse attenuation coefficient at 490 nm taken from the Level-3 binned MODIS-Aqua products in the OceanColor database (available at: http://oceancolor.gsfc.nasa.gov).

The nutrient and salinity thresholds utilised in the coral module are similar to those of ReefHab. The thermal limits however use the temperature in each grid cell at each depth unlike ReefHab which only uses sea surface temperatures.

**2.2.2 Calcium carbonate production**

Once coral habitability has been determined, the production of calcium carbonate ($P$) depends on several local variables (Figure 1). Because the vertical resolution in the model is relatively coarse (increasing from 10 meters at the surface to 28 meters at 100 m depth), coral production is computed on a sublevel vertical grid every meter. The carbonate production is computed as:

$$P = g_{max} \times f_R(PAR) \times f_T(T) \times f_O(\Omega) \times S_{avail} \times TF \times f_B(bleach)$$

Where $g_{max}$ is the maximum value that is a tuning parameter (Table 1), $f_R(PAR)$ a function of the photosynthetically active radiation at the surface ($PAR$), $f_T(T)$ a function of the temperature ($T$), $f_O(\Omega)$ a function of the aragonite saturation state ($\Omega$), $S_{avail}$ the available surface area, $TF$ the topographic factor, and $f_B(bleach)$ a function for the bleaching.

The local variables governing the calcium carbonate production are:

      (a)  Light availability: Calcification is assumed to be directly proportional to  photosynthesis (Chalker, 1981). The

130             production is a function of light depending on surface photosynthetically active radiation ($PAR$) and its attenuation with depth. The function, as for ReefHab, uses a hyperbolic tangent (Jassby and Platt, 1976; Bosscher and Schlager, 1992):


$$f_R(PAR) = \tanh\left(\frac{I_z}{I_k}\right)$$

      where $I_z = PAR \times e^{(-K_{490} \times z)}$ with $z$ the depth at the subgrid level (every meter), $K_{490}$ is again the diffuse attenuation coefficient at 490 nm, and $I_k$ a parameter used in the model tuning (Table 1).





(b) Temperature: the study by Jones et al. (2015) showed that the best results for coral production were obtained with a linear relationship between calcification and temperature. We have thus added a linear function of temperature ($T$), °C, fitted for the temperature range of coral reef habitability ($f_T(T)=0$ at T=18.1°C and $f_T$ (T)=1 at T=31.5°C; $f_T$(T)=0 outside the range of 18.1-31.5°C):

$$f(T) = -1.38 + 0.077 \times T$$

(c) Aragonite saturation state: following Langdon and Atkinson (2005) we have added a function depending on the aragonite saturation state ($\Omega$) defined as the ratio of the ion concentration product to the solubility product ($K_{sp}$) for the mineral aragonite at the in-situ temperature, salinity and pressure:

$$\Omega = \frac{[Ca^{2+}][CO_3^{2-}]}{K_{sp}}$$

The production function is then:

if $\Omega > 1$  $f_O(\Omega) = \frac{\Omega - 1}{K_{arag}}$

Else $f_O(\Omega) = 0$

with $K_{arag}$ the normalisation parameter ($K_{arag}$=2.86).

(d) The available surface area: $S_{avail}$ is computed in each grid cell from GEBCO 2014 (GEBCO Compilation Group, 2022) with a 1 m subgrid vertical resolution. For each vertical 1 m depth interval, we sum the areas from GEBCO corresponding to that level which are contained in a CLIO grid cell. Because of the coarse grid of iLOVECLIM, some ocean areas from GEBCO occur on the continental grid. In which case, the surface area is added to the nearest ocean grid cell. These cases represent very small areas and have negligible impact on model results.

(e) A topographic factor, TF, is used to account for the effect of topography as in ReefHAB. The calculation follows a two-step parametrisation:

1. via a topographic relief for each grid element, denoted $\alpha_{ij}$, derived by summing up the slopes of the lines connecting its midpoint to the midpoints of its eight neighbouring cells:

$$\alpha_{ij} = \sum_{n_i=i-1}^{i+1} \sum_{n_j=j-1}^{j+1} \tan^{-1}\left(\frac{Z_{n_i,n_j} - Z_{ij}}{D_{(n_i,n_j)-(i,j)}}\right)$$

where



$Z_{ij}$ is the depth at the (i, j) midpoint [m];

$Z_{ni,nj}$ is the depth at the $(n_i,n_j)$ midpoint [m];

$D_{(ni,nj)-(i,j)}$ is the distance [m] between midpoints $(n_i,n_j)$ and (i,j)

$\alpha_{ij}$ is furthermore limited to a maximum of 1.7, which appears to be typical of shelf breaks. Atolls would theoretically present a greater relief, but it appears that atolls or reef areas near steeply sloping continental shelves do not accumulate $CaCO_3$ any

faster than shelf break reefs. It should be noted that, the result of $tan^{-1}$ in the equation above needs to be expressed in degrees, in order to reproduce the values of α reported on Fig. 7 from Kleypas (1997).

2. A topography factor, *TF*, was empirically derived from dynamic simulation experiments, focusing on the Great Barrier Reef where actual Holocene accumulation rates are well documented. The effective accumulation rate $G_{eff}$ was then defined as

$$G_{eff} = G \times TF$$

According to Kleypas (1997), the most realistic reef thicknesses are obtained with

$$TF = \frac{ln(\alpha \times 100)}{5}$$

Reefs along outer continental shelves and mid-ocean atolls have *TF* values close to 1.0, while topographically uniform inner shelves have *TF* values near 0.05. α values are limited to a minimum value of 0.01 to avoid physically meaningless negative *TF*s.

(f)    An inhibition function depending on bleaching, detailed below.

The coral carbonate production is computed daily at each subgrid vertical level, i.e. at every 1 m depth interval, for each ocean grid cell within the coral habitability range.

### 2.2.3 Bleaching

Expanding on ReefHab, iCORAL additionally includes a bleaching algorithm based on the degree heating week method used

by NOAA's satellite-based warning system Coral-Reef Watch (https://www.coralreefwatch.noaa.gov/product/5km/methodology.php).

We first compute the maximum of the climatological monthly mean temperature over 30 years, i.e., the temperature of the hottest month in the climatological monthly means relative to the grid element (Maximum of the climatological Monthly Mean temperature $MMM_{clim}$). This climatological reference period can either be fixed to the first 30 years of a simulation, which




corresponds to no bleaching adaptation of corals to changing temperature, or it is continuously updated with a moving 30-year window, to account for some coral adaptation to temperature induced bleaching.

We then compute the degree heating week (*DHW*), an index that determines bleaching if it exceeds a prescribed threshold. *DHW* is a measure of the accumulation of hot spots above 1°C, as prolonged periods of excessive heat are the main driver for bleaching. For this we compute the daily hot spot (*HS*) which is the difference between the daily temperature (*T*) and the $MMM_{clim}$ for the month to which day *j* belongs to:

$$HS_j = T - MMM_{clim}$$

From these daily hotspots, we derive daily excess hotspots, *xHS_j*, defined by

$xHS_j = HS_j$ if $HS_j \geq 1$ and $xHS_j = 0$ otherwise

The *DHW* value for a day *i* is then obtained by summing the daily excess hot spot values over 12 weeks (i.e., 84 days):

$$DHW_i = \sum_{j=i-84}^{i} \left( \frac{xHS_j}{7} \right)$$

The factor of 1/7 is used to convert the final *DHW* to units of degree Celsius-weeks (°C-weeks), as coral bleaching usually develops on the time scale of weeks.

If *DHW* crosses prescribed critical thresholds, it triggers coral bleaching, which then temporarily limits calcium carbonate production: if *DHW* exceeds 4 °C-weeks the bleaching is considered moderate, if *DHW* exceeds 8 °C-weeks it is considered severe.

If bleaching has taken place, coral reef carbonate production is limited by the bleaching according to:

$$f_B(bleach) = 1 - e^{-\frac{t-t_{bleach}}{\tau_{bleach}}}$$

where $t_{bleach}$ denotes the year in which the most recent bleaching event occurred and *t* stands for the current year. If the bleaching is severe, the time constant $\tau_{bleach}$ (used in the computation of future carbonate production limitation) is set to 20 years. If the bleaching is moderate, several cases are considered:

(a) If the coral reef is not currently recovering from a previous bleaching event, the time constant $\tau_{bleach}$ is set to 5 years;

(b) If the coral reef is recovering from a previous moderate bleaching and the time since the previous bleaching event is less than 2 years, then the time constant $\tau_{bleach}$ is set to 20 years (as with for severe bleaching);

(c) If the coral reef is recovering from a moderate bleaching event and the time since last bleaching is greater than 2 years ago, then $\tau_{bleach}$ is unchanged;

(d) If the coral reef is recovering from severe bleaching, $\tau_{bleach}$ is unchanged.

In addition, if the thermal habitability limit (31.5°C) is exceeded, it is also assumed that severe bleaching has taken place ($\tau_{bleach}$=20 years).



If the last bleaching event was sufficiently long ago (4 times the time constant $\tau_{bleach}$, meaning 20 years for a moderate bleaching event and 80 years after a severe bleaching event) coral carbonate production is considered unaffected by bleaching ($f_B(bleach) = 1$).

### 2.2.4 Impact on the carbon cycle

The production of aragonite by coral reefs impacts the carbon cycle by directly modifying the global inventories of DIC [mol/kg] and ALK [eq/kg] in the model:

$$\frac{dDIC}{dt} = -P$$

$$\frac{dALK}{dt} = -2P$$

where $P$ is the global annual carbonate production [mol/kg].

As there was no riverine input of carbon and alkalinity to the ocean in iLOVECLIM by default, we have added a homogenous input of alkalinity and carbon at the ocean surface to represent river inputs from weathering. We consider a global constant value $C_{riv}$= 14 Tmol yr$^{-1}$, assumed to be all in HCO$_3^-$ form, resulting in $A_{riv} = C_{riv}$. This riverine flux is smaller than the actually observed riverine carbon and alkalinity input, because it only compensates for the carbonate loss from the ocean by accumulation in coral reefs, which represents only part of the global ocean carbon and alkalinity sinks.

Weathering removes $CO_2$ from the atmosphere:

$$\frac{dC_A}{dt} = -0.5 \cdot C_{riv}$$

where $C_A$ is the global atmospheric $CO_2$ inventory (PgC).

Note that dissolution of coral-reef carbonates is not yet explicitly included, but will be added in future developments. In addition, we do not consider organic carbon production, but only carbonate production.

### 2.2.5 Temperature variability in iLOVECLIM

Due to its simplified atmospheric module, the temperature variability of iLOVECLIM in the tropics is relatively low compared to observations (Sriver et al., 2014). Unaccounted for, this would bias the simulation of bleaching events using the degree heating weeks method. We have thus generated additional temperature variability in the tropics. For this we use an autoregressive model. Its parameters, including its order, were derived from the analysis of the daily sea surface temperature anomalies in a tropical region with extended coral reef cover (19-16°S, 148-154°E). We fitted a series of AR(p) models (p = 1, …, 6) to the daily time series in each grid point in this area (for details, please see supplementary information) and selected





the AR(1) model, as the RMSE of the higher order models was not statistically different. Accordingly, we generate an AR(1) variate with an auto-correlation parameter of 0.90 and a standard deviation of 0.28 to add daily variability to the otherwise anomalously smooth temperature evolution in iLOVECLIM.

265 **2.3 Simulations**

We have run an ensemble of simulations under pre-industrial boundary conditions (atmospheric $CO_2$ of 284 ppm) with varying values for coral parameters to select the best parameter set compared to existing observational data. To this end, we have run 210 simulations starting from an equilibrium pre-industrial simulation (Bouttes et al., 2015). Since the 2015 version of iLOVECLIM, the pH calculation routine has been replaced by the SolveSAPHE module based upon the $A_{CBW}$ approximation

270 to total alkalinity (Munhoven, 2013). The ensemble of simulations is run with different values for the maximum production parameter $g_{max}$, the saturating light intensity $I_k$ and the minimum light intensity necessary for reef growth $I_{min}$ (Table 1). In these simulations, there is no feedback from the simulated coral reefs to the climate.

| Parameters | Name | Min value | Max value | Step |
|---|---|---|---|---|
| $I_{min}$ (µE m$^{-2}$ s$^{-1}$) | Minimum light intensity necessary for reef growth | 50 | 300 | 50 |
| $I_k$ (µE m$^{-2}$ s$^{-1}$) | Saturation light intensity | 50 | 350 | 50 |
| $g_{max}$ (mm yr$^{-1}$) | Maximum production | 1 | 5 | 1 |

Table 1: Parameter values used in model tuning resulting in an ensemble of 210 simulations. The minimum, maximum and
275 incremental step in parameter values used during model tuning are indicated.

**2.4 Data used to constrain the model**

To constrain the pre-industrial model results we have used published observations of coral reef locations (UNEP-WCMC, 2018; Figure 2), global area, as well as global and regional carbonate production estimates (Perry et al., 2018). Data are mainly

280 for the modern era rather than the pre-industrial. However, a pre-industrial simulation is required in order to initialize historical and future simulations. It is therefore assumed that global coral reef distribution and carbonate production has exhibited limited change over the industrial era. The global area and carbonate production of tropical coral reefs are difficult to evaluate and constrain. According to Vecsei (2004), the total global area ranges between 303 and 345 ×10$^3$ km$^2$ and the global carbonate production between 0.65 and 0.83 Pg CaCO$_3$ yr$^{-1}$. More recent global area estimates indicate a range of 284 ×10$^3$ km$^2$ (Spalding

285 et al., 2001) or 150-300 ×10$^3$ km$^2$ (Li et al., 2020). On the other hand, older studies have suggested larger values ranging from



600 to 1500 $\times 10^3$ km$^2$ (Smith, 1978; Crossland et al., 1991; Copper, 1994). Given this uncertainty and the fact that more recent studies suggest that the largest estimations are probably over estimated, we consider a potential range of 150-600 $\times 10^3$ km$^2$.

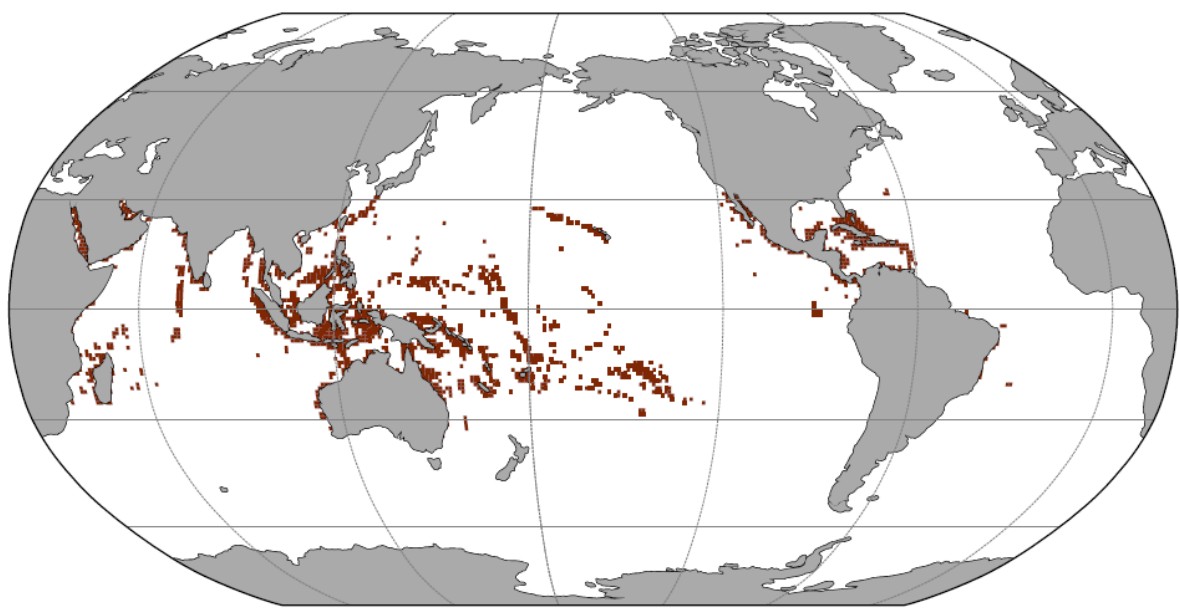

**Figure 2: Coral location from UNEP-WCMC (2018) dataset.**

**3 Results**

We first evaluate the variables simulated by iLOVECLIM that are relevant for coral production, and then compare the coral module results of the ensemble simulations to existing observations of coral reef distribution, area and carbonate production.

**3.1 iLOVECLIM variables**

As described in the methods, the main variables simulated by the model that are used to compute coral reef habitability and production are temperature, salinity, phosphate concentration and aragonite saturation state ($\Omega$) (Figure 3).



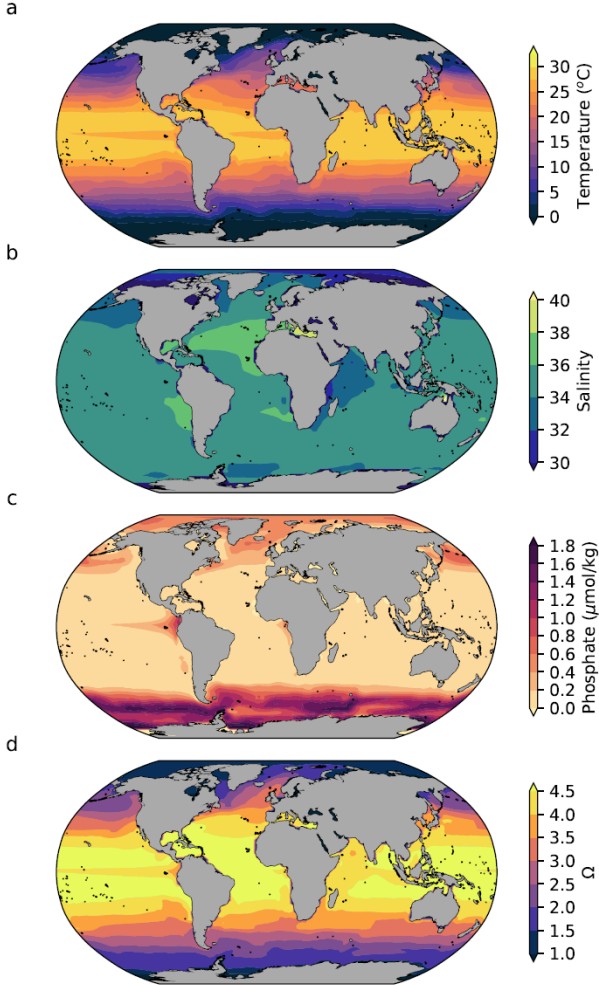

**Figure 3. Surface ocean (a) temperature (°C), (b) salinity, (c) phosphate concentration (µmol kg⁻¹) and (d) aragonite saturation state (Ω) simulated by iLOVECLIM in pre-industrial conditions. The model outputs are 100-year averages at the end of the equilibrium pre-industrial simulation.**

In order to compare the iLOVECLIM variables used for coral reef calcification to modern data, we also consider a historical run following the CMIP protocol (Meinshausen et al., 2020) and average the variables over 2000-2010 (Figure 4). As already evaluated in other studies, iLOVECLIM simulated sea surface temperature and salinity are in general agreement with data (Goosse et al., 2010, Bouttes et al., 2015), albeit with some regional differences, due partly to the relatively coarse resolution of the model (3° horizontally). The distribution of simulated nutrients exhibits greater biases, especially in eastern equatorial



upwelling regions where the concentrations simulated by the model are smaller than observations. The saturation state is also in generally good agreement with data, despite some differences locally.

**Figure 4. Model (left) and observational data (right) surface maps of (a,b) temperature (°C), (c,d) salinity, (e,f) phosphate (µmol/kg) and (g,h) aragonite saturation state (Ω). The model outputs are averaged over 2000-2010. The data are from Locarnini et al. (2018), Zweng et al. (2018), Garcia et al. (2018) and Jiang (2015).**




## 3.2 Location and global reef area

The location of simulated tropical reefs is globally in broad agreement with observational data (Figure 5). The model computes the presence of corals in most locations where coral reefs have been observed (in blue). However, the model tends to overpredict coral development, i.e., simulates corals in regions where they are not observed, notably in the Atlantic basin (in beige and purple). It furthermore fails to simulate some coral locations observed in data (in brown), but this mismatch is less widespread. The model could predict coral presence in places where it has not yet been observed, but the overprediction might

also be due to the lack of rivers in the model. Indeed, high nutrient concentrations typically prevent coral reef development due to competition with macroalgae, and in coastal regions high nutrient concentrations can be partly due to riverine inputs which are not represented in the model. This could explain some of the mismatch west of Africa. In addition, the model also simulated small isolated coral reefs with small areas (in purple) that might not be present in the observed data.

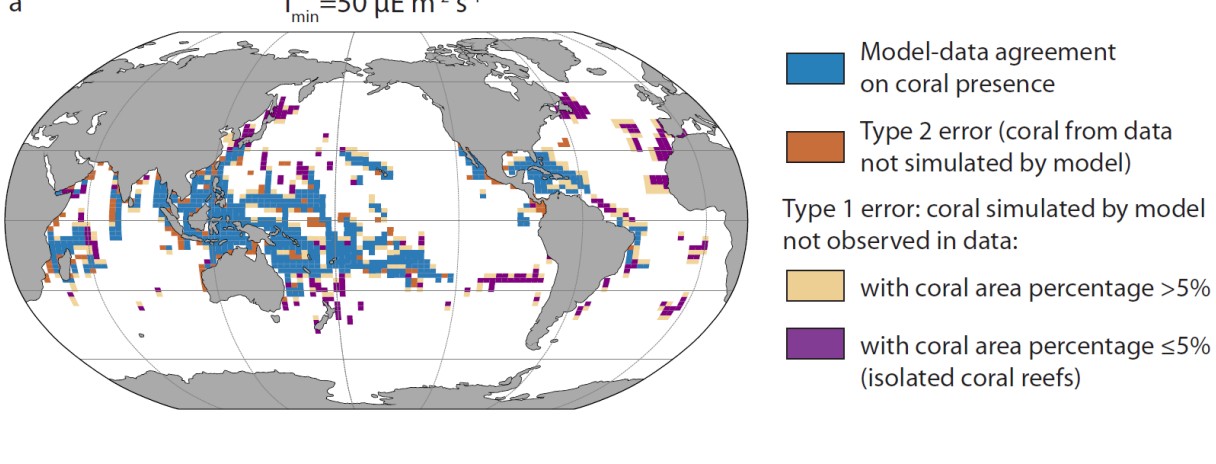

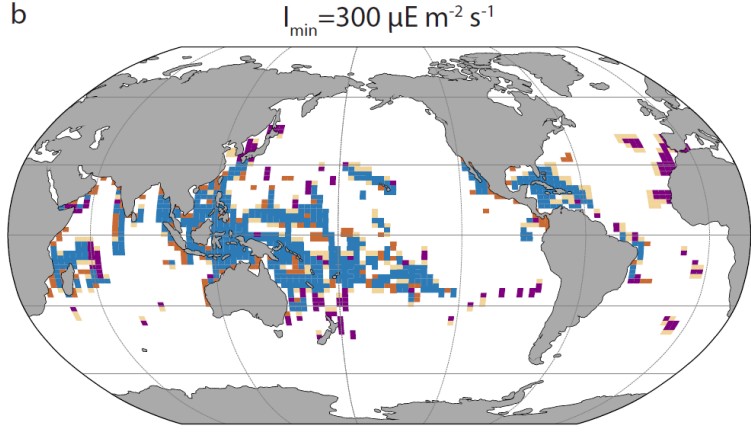

**Figure 5. Coral location in the model and data for the minimum and maximum $I_{min}$ (the minimum light intensity necessary for reef growth) values. Blue cells indicate the presence of corals in both model and observational data, brown cells indicate the presence of corals in observational data but not in the model simulation, beige and purple indicate the presence of corals in the model simulation**





**but not in observations. Some locations correspond to places with very small surface areas (purple, due to small islands for example)**
**but as we plot the presence of corals in the relatively large oceanic grid cells (the horizontal ocean resolution is 3°x3°) it might give**
**the impression of large coral coverage.**

| $I_{min}$ ($\mu$E m$^{-2}$ s$^{-1}$) | Model-data agreement | Type 2 (false negative) error | Type 1 (false positive) error, excluding isolated corals | Type 1 (false positive) error, only isolated corals |
|---|---|---|---|---|
| 50 | 595 | 159 | 238 | 226 |
| 300 | 576 | 178 | 170 | 154 |

**Table 2. Number of model grid points with model-data agreement or disagreement. The isolated coral reefs are defined when coral**
**area ≤ 5% of the total area between 0 and -50 m (last column).**

The global coral reef area depends on the simulated habitability, which is set by local environmental variables computed by the model, i.e. temperature, salinity and nutrients, which are identical across our simulations as they are independent of coral carbonate production. It also depends on light availability, and attenuation with depth. The minimum light intensity needed for

coral growth is set by the $I_{min}$ parameter that is changed in our simulation ensemble. Hence $I_{min}$ is the only parameter among the varied parameters and functions that impacts the simulated reef area.

As $I_{min}$ increases the critical depth down to which sufficient light penetrates becomes shallower, and as a result, the global area covered by coral reefs decreases. The total area ranges from 1500 $\times 10^3$ km$^2$ with $I_{min}$=50 $\mu$E m$^{-2}$ s$^{-1}$ to 390 $\times 10^3$ km$^2$ with $I_{min}$=300 $\mu$E m$^{-2}$ s$^{-1}$ (Table 2). This is less than in Kleypas (1997) for the same $I_{min}$ parameter values, and in better agreement

with observational data, but still high compared to the observed range of 150-600$\times 10^3$ km$^2$ (Vecsei, 2004; Li et al., 2020) for most simulations. The low range total areas are nonetheless in agreement with three other model estimations computed by Jones et al. (2015) with the KAG (492 $\times 10^3$ km$^2$), LOUGH (567 $\times 10^3$ km$^2$) and SILCCE (500 $\times 10^3$ km$^2$) models. The total coral reef area is very uncertain, and there are possibilities of both under estimation by data and overprediction by the model.



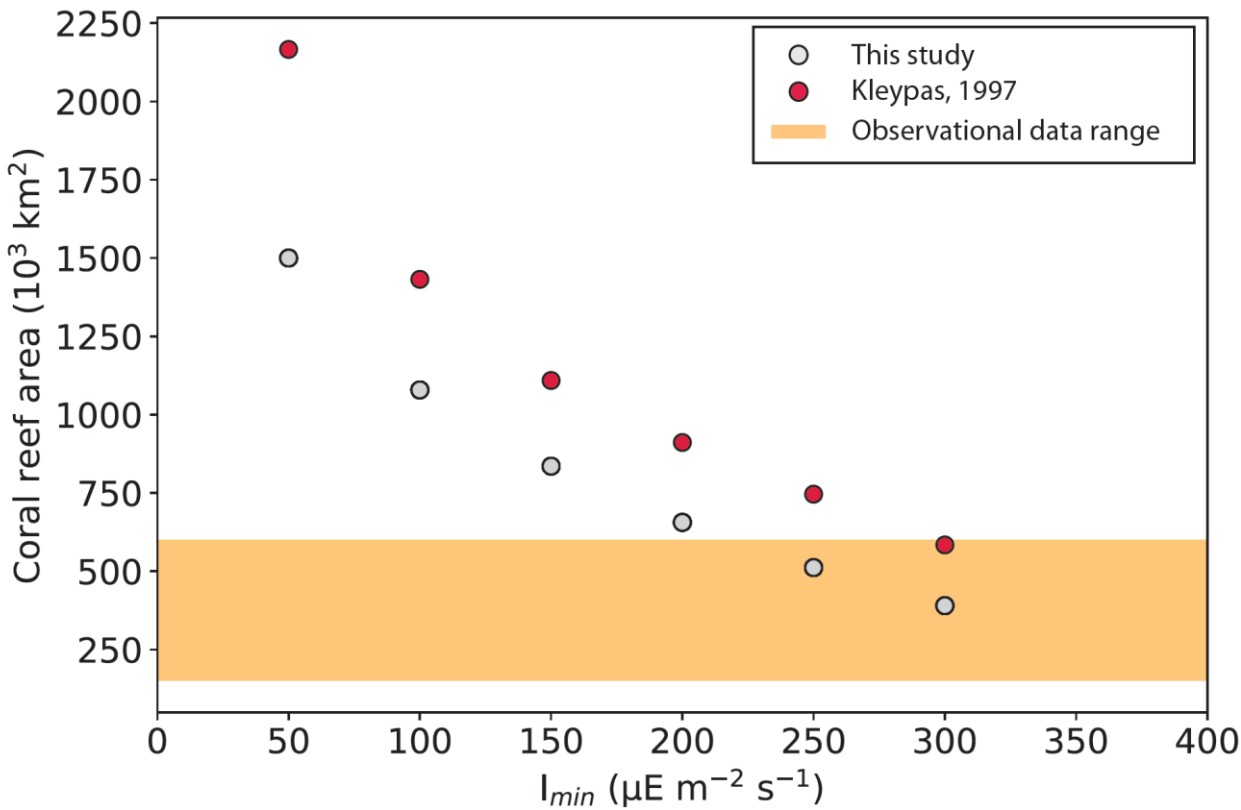


**Figure 6. Global coral reef area (10³ km²) simulated in this study and in Kleypas (1997) as a function of $I_{min}$ (the minimum light intensity necessary for reef growth, µE m$^{-2}$ s$^{-1}$). The range of observational data for the global coral reef area (section 2.4) is shown with the yellow bar.**

**3.3 Global and regional calcium carbonate production**

According to observation-based estimates, global coral reef carbonate production is between 0.65 and 0.83 Pg CaCO$_3$ yr$^{-1}$ (Vecsei, 2004). In our ensemble of simulations, global carbonate production ranges from 0.27 to 8.84 Pg CaCO$_3$ yr$^{-1}$. Simulations with global production within the observational range can be found for all $I_{min}$ and $I_k$ values, but only for $g_{max}$ from 1 to 3 mm yr$^{-1}$ (Figure 7). The largest global production is obtained for the lowest $I_{min}$ and $I_k$ values of 50 µE m$^{-2}$ s$^{-1}$, when the

light limitation is less stringent. The largest production is also obtained for the largest $g_{max}$ (maximum production parameter) value of 5 mm day$^{-1}$. Contrary to this, low production is obtained with high $I_{min}$ and $I_k$, and low $g_{max}$.







**Figure 7. Global coral reef carbonate production (Pg CaCO₃ yr⁻¹) as a function of (a) $I_{min}$ (the minimum light intensity necessary for**
**reef growth, µE m⁻² s⁻¹), (b) $I_k$ (the saturating light intensity, µE m⁻² s⁻¹) and (c) $g_{max}$ (the maximum production growth). The range**
**of observational data for the global carbonate production (Vecsei, 2004) is shown with the yellow bar.**

When considering model performance with regards to both global reef area and global carbonate production, only six
simulations display values in the range of observation-based estimates (Figure 8 and Table 3). The main limitation comes from
the coral reef area, as most of the simulations overestimate coral reef area, with only a handful located within the observed
values (Figure 6).

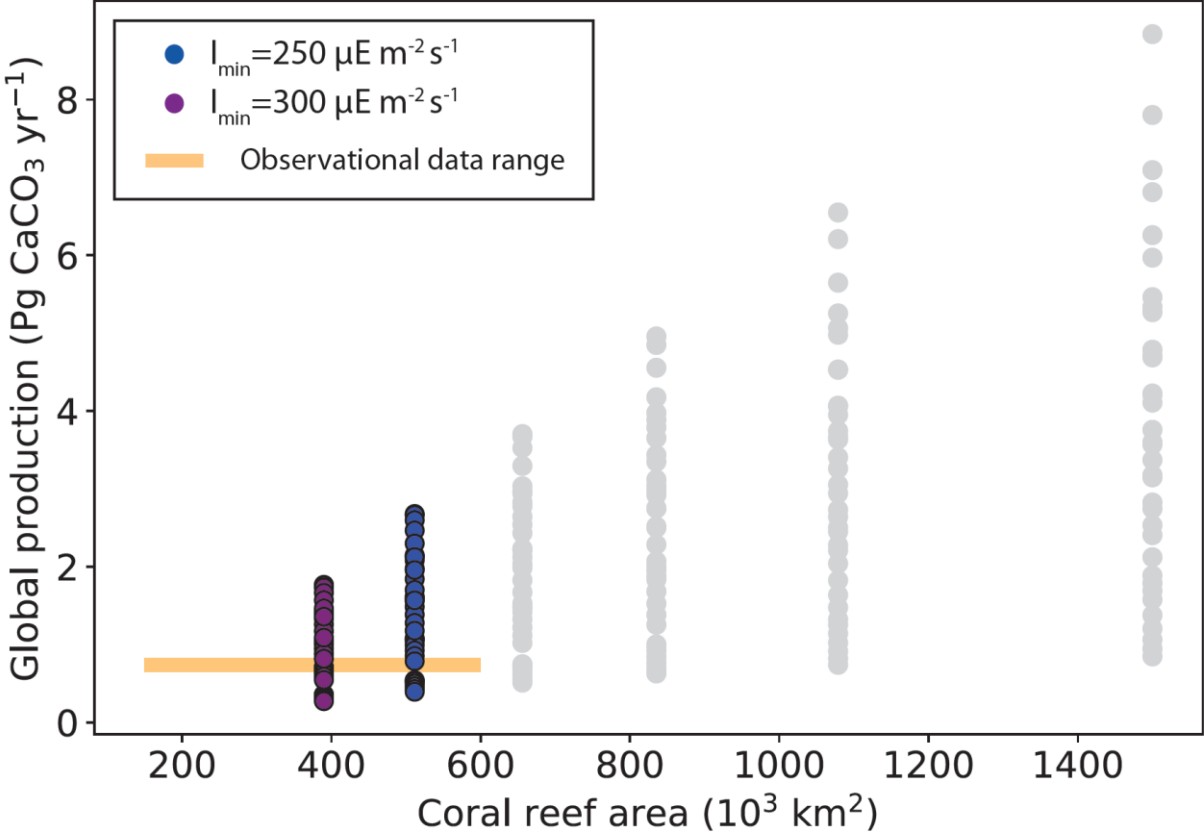

**Figure 8. Global carbonate production (Pg CaCO₃ yr⁻¹) as a function of global coral reef area (10³ km²).**





| $I_{min}$ (µE m$^{-2}$ s$^{-1}$) | $I_k$ (µE m$^{-2}$ s$^{-1}$) | $g_{max}$ (mm yr$^{-1}$) | Global reef area (10$^3$ km$^2$) | Global Production (Pg CaCO$_3$ yr$^{-1}$) | Regional RMSE (kg CaCO$_3$ m$^{-2}$ yr$^{-1}$) |
|---|---|---|---|---|---|
| *DATA* | | | 150-600 | 0.65-0.83 | |
| 250 | 350 | 2 | 512 | 0.79 | 2.01 |
| 300 | 50 | 2 | 390 | 0.71 | 1.90 |
| 300 | 100 | 2 | 390 | 0.71 | 1.90 |
| 300 | 150 | 2 | 390 | 0.70 | 1.91 |
| 300 | 200 | 2 | 390 | 0.67 | 1.95 |
| 300 | 350 | 3 | 390 | 0.82 | 1.81 |

**Table 3. Global carbonate production, tuning parameters and root mean square error relative to regional production data (Perry et al., 2018) for the simulations with both global production and total area within observational constraints.**

We finally compare model results with the regional carbonate production data from Perry et al. (2018). Figure 9 shows the root mean square error (RMSE) between model results and observational data for regional carbonate productivity, as a function 380 of global production or global coral reef area. Depending on the parameter choices (Table 3), the model-data agreement varies greatly. The simulations in agreement with both global production and coral reef data are also among those with the lowest regional production RMSE (Table 3), ranging from 1.81 to 2.01 kg CaCO$_3$ m$^{-2}$ yr$^{-1}$, hence in better agreement with all observed data.

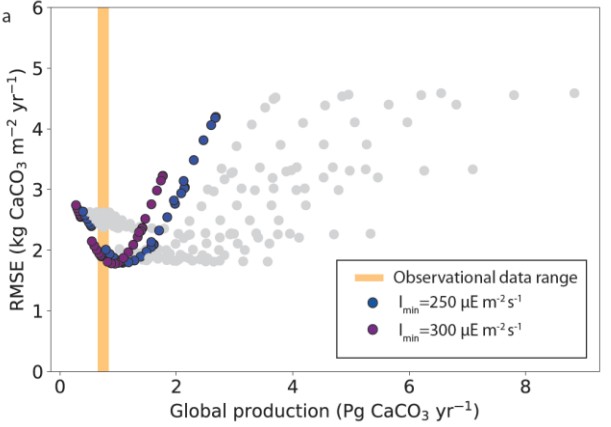

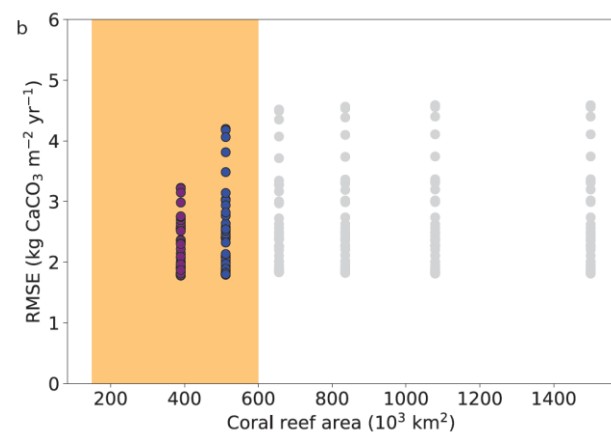




**Figure 9. Root mean square error (RMSE, kg CaCO₃ m⁻² yr⁻¹) between the simulations and the observational data of regional production (Perry et al., 2018) as a function of (a) global production (Pg CaCO₃ yr⁻¹) or (b) coral reef area (10³ km²).**

The six best performing ensemble simulations when considering both regional and global observational constraints are given in Table 3. All these ensemble members simulate global production within the range of data-based estimates. In these simulations, we have selected the ensemble member with the lowest RMSE (hence closest agreement with regional production data). Our optimal parameter choices are therefore $I_{min}$=300 μE m⁻² s⁻¹, $I_k$=350 μE m⁻² s⁻¹ and $g_{max}$= 3 mm yr⁻¹. For this simulation, the global carbonate production is 0.82 Pg CaCO₃ yr⁻¹ and the global coral reef area is 390×10³ km².

## 4 Discussion

We have presented a new module to compute coral reef production and integrated this module in the iLOVECLIM carbon-climate model. Contrary to Jones et al. (2015), where the coral reef modules were forced by climatic data, we have embedded our module in the coupled carbon-climate iLOVECLIM model. While this will be particularly useful to evaluate coral-climate carbon cycle feedbacks and the response of corals to climate change, it also entails that the module performance will be influenced by the model biases.

### 4.1 Model caveats

The first limitation is due to the model resolution. The ocean component of iLOVECLIM is a full GCM with 3° horizontal resolution and 20 vertical levels. Hence local scale changes of temperature, saturation state, or light penetration below 3° cannot be accounted for in our model. Future work should therefore evaluate the performance of the coral module within higher resolution ocean components. The vertical resolution limitation is partly resolved through the use of a subgrid vertical scale of 1 meter to account for light attenuation, but temperature and aragonite saturation state are uniform in each grid box, for which a higher resolution model would also be useful.

In addition, the simulated nutrient distribution of iLOVECLIM is locally different from observational data. In particular, there are no riverine inputs in iLOVECLIM, resulting in a lack of enhanced nutrient concentrations near river mouths, which can influence coral habitability. This could be more closely looked at in models including rivers input. Finally, light attenuation in the model is currently prescribed based on satellite data. Ideally however, it would take into account simulated phytoplankton biomass and be computed using marine productivity.



## 4.2 Future developments

Besides the climate model, other limitations come from the coral reef module itself. In terms of coral representation, we have only one type of coral representing all communities. However, different communities (or species) respond differently to the driving variables such as temperature (Coles and Brown, 2003; D'angelo et al., 2015) and aragonite saturation state (Chan and Connolly, 2013; Kroeker et al., 2010; Kroeker et al., 2013). Further development could thus include several communities with different parameters for the temperature and omega function for each of them, similar to what is done for plankton and

zooplankton, or for plant functional types (PFTs) on land.

Adaptation to temperature changes is currently an option in the module. The computation of the Maximum of the climatological Monthly Mean temperature ($MMM_{clim}$) can either be set to the first 30 years of the simulation (no adaptation) or be set to a rolling mean over a 30-year window evolving in time (adaptation). While adaptation is potentially crucial for coral reefs (Logan et al., 2021), its quantification is poorly constrained, and would require more work. In addition to some form of adaptation to bleaching, adaptation of the thermal habitability range could also be taken into consideration. If different coral communities

are considered in the future, adaptation could also depend on the coral community.

Dissolution is not yet included, as no existing modern data would allow us to validate this part of the module, but future work considering coral reefs in the past will implement it and use past coral evolution to validate this new addition.

Some processes such as erosion and bioerosion (Schönberg et al., 2017) are also not currently considered, as they are likely to

be of second order, or are insufficiently constrained to be included at this stage. In the future, as more knowledge is gathered, they might be worth adding in the module.

The sea level rise due to global warming will make more coastal area potentially available for the coral reef growth. This effect could be captured with a parameterization of coral growth dependence on a rate of sea level change (Munhoven and François, 1996; Kleinen et al., 2016).


## 4.3 Observational constraints on model development

Finally, the model representation depends highly on the functions of environmental variables. The only way to improve this part is through more constraints from in situ and laboratory experiments yielding more information on the functions and parameters used in the model. This modelling approach will thus benefit from all future studies focusing on the response of

coral reefs to the values of environmental variables such as temperature or the saturation state.



## 5 Conclusions

In conclusion, we have developed a new module, called iCORAL, of coral reef aragonite calcification based on ReefHab (Kleypas, 1995, 1997) for usage in Earth System Models. The new developments account for the role of temperature and the

saturation state with respect to aragonite in the carbonate production rate. We have furthermore added a simple bleaching scheme based upon the successful NOAA Coral Reef Watch rationale. iCORAL has been implemented in the climate-carbon model of intermediate complexity iLOVECLIM. The simulations with iCORAL-iLOVECLIM are in fair agreement with data in terms of total productivity and areal distribution, as well regional productivity. iCORAL-iLOVECLIM is ready to use for studies of coral reef changes in future and past periods, when the role and feedbacks of shelf carbonate accumulation rate

changes on the carbon cycle (and hence on climate) need to be evaluated.

**Code availability**: The code of the iCORAL module is available on Zenodo (doi: 10.5281/zenodo.7985881).

**Data availability:** The simulation outputs will be available on Zenodo.


**Author contribution:** NB and GM developed the model code. NB, GM and LK have designed the simulations, NB and MB performed them. NB prepared the manuscript with contributions from all co-authors.

**Competing interests**: The contact author has declared that none of the authors has any competing interests


**Acknowledgments:** We thank Olivier Torres for helping with coral data processing. Financial support for this work was provided by the Belgian Fund for Scientific Research – F.R.S.-FNRS (project SERENATA, grant no. CDR J.0123.19). Guy Munhoven is a Research Associate with the Belgian Fund for Scientific Research – F.R.S.-FNRS.

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
