# Peer review of "Implementing the iCORAL (version 1.0) coral reef CaCO3 production module in the iLOVECLIM climate model"

_EGUsphere, 2023_

## Referee Comment (RC2)

**Review of:**

**Implementing a coral reef CaCO3 production module in the iLOVECLIM climate model**

*by Nathaelle Bouttes and co-authors / egusphere-2023-1162*

01/17/2024

Coral reefs and associated controls on carbonate precipitation and burial and a potentially key, but to date overlooked, interactive carbon cycle component in Earth system models. To my knowledge the authors are right in that previously, only prescribed, rather than interactive, carbon and alkalinity sinks associated with shallow carbonate production have been implemented in (3D ocean-based) global models. As such, the current work represents a very useful modeling advance and highly appropriate for the time-scale capability of the 'iLOVECLIM' climate (Earth system?) model. The paper is well-written and the model parameterizations generally well described and justified. I do have a number of minor comments (listed later). However, I do also have some questions about whether some of the assumptions made in the construction of the coral reef CaCO3 production module tie carbon and alkalinity feedback too closely to the modern marine environment and observations, preventing direct past (geological) applicability and potentially also somewhat limiting future capabilities of the new coupled model.

- **Generalizability/applicability of the model**

  A couple of assumptions are made in the coral reef CaCO3 production module have implications for its applicability to non-modern, and particularly paleo situations.

1. Diffusive attenuation coefficient

   As an initial note – I think URLs are not allowed these days(?) I did go to the page and try and retrieve the data, but either I was being incompetent, or the details given in the text are insufficient to retrieve the specific data in question. Ideally, the retrieved data would be placed somewhere with a

DOI. I did check the DOI given for iCORAL, but it is only the FORTRAN file and does not include any boundary conditions. A DOI is in any case needed for the current version of iLOVECLIM, and that could then include relevant boundary conditions such as for the K490 field(?)

So my question is: how important is the diffusive attenuation coefficient field? If a mean global value was applied uniformly, or representative open ocean value applied uniformly, how different does the projected distribution of reefs and global carbonate production become?

Using the present-day satellite-retried spatial pattern potentially strongly pins the modelled distribution to 'now'. In the future with changing river flow, sediment loads, etc., the pattern may change, introducing a bias in future simulations. Much more problematic would be paleo applications, particularly when the land-sea mask is different and one can no long map present-day satellite retrievals onto past oceans. What are the authors plans for applying iCORAL-iLOVECLIM to the geological past and what are they planning to assume re. K490?

My guess, given that the baseline model (Figure 5a) struggles with e.g. correctly projecting the absence of reefs in the NW Atlantic anyway, is that high sediment loads and the absence of hard substrates may be more important than getting light attenuation 'right'. Hence I wonder whether one could apply a mean or representative value globally, accept a small degradation in model fidelity, but remove this tie to the present-day?

2. Sea-floor bathymetry

I understand exactly why the authors have imposed a much higher resolution sea-floor bathymetry on the reef module. However, while for far future simulations one could simply take into account a mean sea-level change, things become (isostatically) more complicated if you go back to e.g. the last glacial maximum, and I am sure that (and the glacial-interglacial cycles in general) will be a scientific target of the authors.

If the paleo questions were restricted to the last glacial cycle, then relatively high resolution (10 minute) reconstructions are available, e.g. ICE-6G-C, GLAC-1D, as per PMIP4. What would the coral reef coverage and carbonate production look like if the 0 ka dataset from ICE-6G-C was used? If the authors plan deglacial (ICE-6G-C) or penultimate deglaciation (GLAC-1D) applications, if would be worth-while in the current paper calibrating and evaluating a slightly lower-resolution pale-enabled version of iCORAL/iLOVECLIME using e.g. ICE-6G-C bathymetry data.

Moreover, I cannot help but wonder what the results of simply using the iLOVECLIM ocean grid would be. Sure, reef locations would be very patchy, but as long as there was some sort of distribution of reef occurrence between Pacific, Indian, and Atlantic Ocean basins, I see no reason why the feedback between climate and carbonate removal should not be equally plausible (given a tuning resulting in a plausible initial global carbonate burial flux). This would make iCORAL-iLOVECLIM generically (and equally) applicable past global carbon cycle/climate questions.

As an aside – I did not see the bathymetric resolution stated. The authors state that they bathymetry comes from 'GEBCO 2014' and cite 'GEBCO Compilation Group, 2022'. Going to the GEBCO Compilation Group website, the current data-set is 2023 and at a resolution of 15 arc-seconds. No dataset further back than 2019 is available that I could see and so I am unsure what 'GEBCO 2014' refers to. So a little more detail on the dataset used is needed.

3. Temperature variability in iLOVECLIM

The 3rd assumption that ties iCORAL to the present-day is the imposition of enhanced sea-surface temperature variance in the tropics. Again,

I can see the reasoning behind this, but some details are missing. In particular, the text says: 'for details, please see supplementary information' but I could not find the SI anywhere.

How big an effect is this? Is is a relatively small effect, or is it fundamental to getting the distribution of reefs and global carbonate sink anywhere near correct (a comparison would be helpful to see and I suspect informative to readers)? If the former – could not the bias imposed by adopting un-adjusted iLOVECLIME climatology be 'tuned away'? If the latter – what confidence do the authors have in future and past applications? I was under the impression that variance may change in the future. If only a little, then this may not matter. But what about the last glacial, or the Eemian? Would SST varience be expected to be more, less, or about the same? i.e. how safe is the assumption of observationally-derived SST variance in the past?

Lastly, why only restrict the modification to the tropics? Why not globally? I guess one answer is that there are very few reefs outside of $\pm30°$ (Figure 2). However, rather more model-projected reef locations occur outside of the tropics (Figure 5a,b), and there will potentially be a very different (and spurious) bleaching response either side of the boundary.

I think in general and across all this points above – firstly, knowing the importance (or not) of making the various assumptions and imposing boundary conditions derived from modern observations would be informative and helpful. Secondly, the more that iCORAL can utilized internal iLOVECLIM fields and boundary conditions, the more generally applicable it will be to the future and particularly the geological past. If the authors do not want to make the choice between more 'realistic' and modern-tied vs. a poorer fidelity simulations of present-day reef distributions and global carbonate productions, then why not calibrated, evaluate, and present, two (or more) alternative setups and calibrations that could be used with iLOVECLIM applied to different questions? Overall, many of the choices and assumptions made in developing iCORAL seem to be orientated towards reproducing observations rather than enabling carbon-climate feedback and the stated aim of 'past and future

*coral-climate coupling'*.

– **Model fields and coral reef location evaluation**

I think missing is a sufficiently critical discussion of the model fields driving iCORAL (Figure 4). To me, the surface ocean saturation is rather lower in the tropics in iLOVECLIM vs. observations, while nutrients – which are assumed to prevent reef formation above a threshold – are higher. (Note that the depth of the 'surface' layer is not given in the figure and needs to be.) There are more localized mismatches in temperature and salinity which may or may not also play a role.

I am not at all concerned about the existence of model-data mismatches, which is par for the course, but rather that their potential implications are not sufficiently discussed. 3 parameters are tuned and I wonder to what degree they are countering errors in the simulated environmental fields. In all biogeochemical modeling of this sort, the risk is always that you correct for a deficiency by distorting something else, with the potential that e.g. the strength of carbonate-climate feedback could end up very different.

I think that at the very least, more discussion about how biases in certain simulated environmental parameters and regions might impact projected reef locations. Further evaluating iCORAL by feeding it observed fields (Figure 4a) in place of simulated fields (Figure 4b) would be interesting. Replacing fields one-by-one might be further instructive. One could do this comprehensively, potentially even re-tuning iCORAL for each combination of simulated or observed environmental fields. Or it might be sufficient in the paper simply to take iCORAL as it is (and its current tuning), and test swapping out the simulated for observed fields.

In addition, there may be better ways of comparing simulated and observed fields (Figure 5). For instance, for each observed reef location, one could pull out the simulated and observed values at those locations and cross-plot, perhaps color-coding for basin. Or color-code as per in Figure 6 and pulling out both 'real' and simulated locations. This would be a way to try and identify whether there are any specific model environmental biases which tended to generate false positives or negatives in reef location.

The more you can pull out specifically why – in terms of simulated environmental conditions or model parameters or structure – false positives or negatives occur, the more we'll learn.

- **Minor comments**[1]

– **18** – 'feedback' would be a better (much more commmon) word than '*retroaction*'.

– **24-25** – '*The model enables assessment of past and future coral-climate coupling on seasonal to millennial timescales*' – just noting the aim in the context of the present-day assumptions and my comments above.

– **52-53** – You don't have to add them, but just pointing out some empirical / machine learning papers: Couce et al., Future habitat suitability for coral reef ecosystems under global warming and ocean acidification, Global Change Biology DOI: 10.1111/gcb.12335 (2013); Couce et al., Tropical coral reef habitat in a geoengineered, high-CO2 world, GRL 40, doi:10.1002/grl.50340 (2013).

– **87-88** – Without digging out Millero (1995), I can't remember whether it included anything about solving the carbonate system or not. If not, missing are details of the numerics. Millero (1995) is also full of typos in various dissociation constant coefficients, so there must be a better reference for what the authors have adopted in terms of e.g. dissociation constants.

– **89** – '*nitrates*'? Do you mean: nitrate and ammonia. Or nitrate and ammonia and dissolved N2? Or just NO3, which would be singular 'nitrate'?

– **107-109** – Please add a brief justification for the limits. e.g. I think the northern Red Sea reaches 41 PSU around the Gulf of Suez (Google further tells me that there are 35 coral taxa in the Gulf of Suez). For phosphate – is this a real-world threshold, or chosen in light of the iLOVECLIM surface nutrient simulation? Looking at the WOA annual mean surface PO4, locations incorrectly simulated in the model in Figure 5a lie in surface waters with PO4 above 0.2 – here I am looking at the NW Atlantic and SE Pacific. Is plankton productivity (and turbidity) not the more proximal factor
* * *
[1]Suggested text changes indicated with → and suggested inserted words underlined. **x** represents line number.

influencing the presence/absence of corals (with nutrient availability influencing productivity)?

I noticed that only later down the text does it state: '*The nutrient and salinity thresholds utilized in the coral module are similar to those of ReefHab.*' It would still be helpful to know a little more on the justification, and how important these assumptions are in leading to e.g. Figure 5a.

– **Section 2.2.2** – A schematic of the gridding and grid relationships would be helpful. Maybe pick a single illustrative region and show of he grid relate, both horizontally and vertically. This could be combined with Figure 1 as a second panel (or thrown in SI).

– **150** – $K_{arag}$ could be confused with $K_{sp}$ (of aragonite) to a sloppy reader like myself. If it is a saturation value (reference value or threshold), why not $\Omega_{ref}$ or something?

– **164-165** – Text describing the relationship between grids, gradients, etc. would be much clearer with a figure (see earlier comment).

– **196** – It is a shame there there is not a DOI or anything less nebulous than a webpage ('*https://www.coralreefwatch.noaa.gov/ product/5km/methodology.php*').

– **Section 2.2.4** – I don't know why this doesn't come across clearer. It is correct (in terms of DIC and ALK relationships and flux balance), but a little round-about.

– **Section 2.2.5** – See comment on present-day observationally-tied temperature variance.

– **269-270** – Maybe make this clearer earlier in the text (see earlier comment).

– **452** Given iCORAL is embedded within iLOVE-CLIM, we need a DOI for the version of iLOVE-CLIM used (indeed, the code for iCORAL utilizes a number of iLOVECLIM modules and the iCORAL code is insufficient in isolation).

---

## Author Comment (AC4)

**Autoregressive Model to Parametrise Temperature Variability**

Guy Munhoven

20th September 2021

**Abstract**

A series of autoregressive models is used to analyse sea-surface temperature time series in order to derive a simple parametrisation of temperature variability in a climate model.

**Introduction**

**1 Data**

- Origin: `https://iridl.ldeo.columbia.edu/SOURCES/.NOAA/.NCDC/ .OISST/.version2/.AVHRR/.anom/lat/%2819S%29%2816S%29RANGEEDGES/ T/%281%20Jan%201982%29%2831%20Dec%202015%29RANGEEDGES/lon/%28148E% 29%28154E%29RANGEEDGES/data.nc`

- time: 1982-01-01 to 2015-12-31 (12418 days)

- lat: $-18.875$ to $-16.125$ (12 latitudes at $0.25°$ resolution)

- lon: $148.125$ to $153.875$ (24 longitudes at $0.25°$ resolution)

**2 Procedure**

We fit a sequence of AR($p$) models to the time series extracted from the dataset at each lon-lat grid point, with $p = 1, \ldots, 6$ in order to determine the optimum order:

$$y_t = c + \phi_1 y_{t-1} + \ldots + \phi_p y_{t-p} + \epsilon_t$$

where $c$ is a constant, $\phi_1, \ldots, \phi_p$ are the $p$ parameters of the model and $\epsilon_t$ is white noise (with zero mean and standard deviation $\sigma_\epsilon$).

The time series are first filtered using a Fast Fourier Transform (FFT) method with a high-pass triangular frequency filter in order to get rid of the seasonal cycle and any possible long-term trends. The cut-off frequency is set to 1.25 cycles/year. Since the total number of data points in each time series involves a comparatively large prime factor ($12418 = 2 \cdot 7 \cdot 887$), 130 points were dropped to reduce the time series length to $12288 = 2^{12} \cdot 3$, which allows for more efficient usage of the FFT algorithm.

**3   Results**

Each data series has a zero mean and therefore $c = 0$ for each model. We therefore fit an AR($p$) model

$$y_t = \phi_1 y_{t-1} + \ldots + \phi_p y_{t-p} + \sigma_\epsilon \upsilon_t$$

where $\upsilon_t$ is white noise with a standard deviation of 1. We fit such a model at each lon-lat point in the domain. Below, we report the averages of the parameter values $\phi_1, \ldots, \phi_p$ and of the required $\sigma_\epsilon$, together with the standard deviations of their distributions.

**3.1   AR(1)**

$$y_t = \phi_1 y_{t-1} + \sigma_\epsilon \upsilon_t$$

$$
\begin{aligned}
\phi_1 &= 0.8964 \pm 0.0060 \\
\sigma_\epsilon &= 0.2758 \pm 0.0094
\end{aligned}
$$

**3.2   AR(2)**

$$y_t = \phi_1 y_{t-1} + \phi_2 y_{t-2} + \sigma_\epsilon \upsilon_t$$

$$
\begin{aligned}
\phi_1 &= 1.0873 \pm 0.0097 \\
\phi_2 &= -0.21230 \pm 0.0069 \\
\sigma_\epsilon &= 0.2694 \pm 0.0091
\end{aligned}
$$

**3.3 AR(3)**

$$y_t = \phi_1 y_{t-1} + \phi_2 y_{t-2} + \phi_3 y_{t-3} + \sigma_\epsilon v_t$$

$$
\begin{aligned}
\phi_1 &= \phantom{-}1.115 \pm 0.010 \\
\phi_2 &= -0.354 \pm 0.015 \\
\phi_3 &= \phantom{-}0.130 \pm 0.011 \\
\sigma_\epsilon &= \phantom{-}0.2671 \pm 0.0089
\end{aligned}
$$

**3.4 AR(4)**

$$y_t = \phi_1 y_{t-1} + \phi_2 y_{t-2} + \phi_3 y_{t-3} + \phi_4 y_{t-4} + \sigma_\epsilon v_t$$

$$
\begin{aligned}
\phi_1 &= \phantom{-}1.112 \pm 0.011 \\
\phi_2 &= -0.345 \pm 0.015 \\
\phi_3 &= \phantom{-}0.101 \pm 0.015 \\
\phi_4 &= \phantom{-}0.0251 \pm 0.0077 \\
\sigma_\epsilon &= \phantom{-}0.2670 \pm 0.0088
\end{aligned}
$$

**3.5 AR(5)**

$$y_t = \phi_1 y_{t-1} + \phi_2 y_{t-2} + \phi_3 y_{t-3} + \phi_4 y_{t-4} + \phi_5 y_{t-5} + \sigma_\epsilon v_t$$

$$
\begin{aligned}
\phi_1 &= \phantom{-}1.111 \pm 0.011 \\
\phi_2 &= -0.348 \pm 0.016 \\
\phi_3 &= \phantom{-}0.112 \pm 0.015 \\
\phi_4 &= -0.009 \pm 0.011 \\
\phi_5 &= \phantom{-}0.0308 \pm 0.0062 \\
\sigma_\epsilon &= \phantom{-}0.2669 \pm 0.0088
\end{aligned}
$$

**3.6 AR(6)**

$$y_t = \phi_1 y_{t-1} + \phi_2 y_{t-2} + \phi_3 y_{t-3} + \phi_4 y_{t-4} + \phi_5 y_{t-5} + \phi_6 y_{t-6} + \sigma_\epsilon v_t$$

$$\begin{aligned}
\phi_1 &= 1.110 \pm 0.011 \\
\phi_2 &= -0.348 \pm 0.015 \\
\phi_3 &= 0.109 \pm 0.015 \\
\phi_4 &= 0.001 \pm 0.012 \\
\phi_5 &= -0.003 \pm 0.012 \\
\phi_6 &= 0.0300 \pm 0.0098 \\
\sigma_\epsilon &= 0.2668 \pm 0.0088
\end{aligned}$$

**4   Discussion**

The differences between the performances of the models of subsequent orders are generally small: $\sigma_\epsilon$, which is also the root mean square error of the model reduces by 2.2% from AR(1) to AR(2), by another 0.88% from AR(2) to AR(3), and by only 0.03% from AR(3) to AR(4).

Accordingly, there is little to no justification in calling upon a more complex model than AR(3).

Tests using a normally distributed random series with zero mean and unit variance for $\upsilon_t$ indicate that the temperature distributions in the time series and in the generated AR($p$) model series are very similar for $p = 1, \ldots, 6$. As a consequence, even the AR(1) model might be sufficient.

---

## Author Comment (AC5)

**Reply to RC1**: 'Comment on egusphere-2023-1162', Anonymous Referee #1, 03 Nov 2023

Review: Implementing a coral reef CaCO3 production module in the iLOVECLIM climate model

Bouttes et al.

**

GENERAL COMMENTS

**

This paper presents a new model of coral reef carbonate production, designed to be coupled to earth system model frameworks and allowing for the simulation of long-term feedbacks between coral calcification and the larger carbon cycle. I am not aware of any other models of coral calcification that are designed to be run in a fully-coupled manner with ESMs, and as such agree that this model fills a previously unfilled niche in climate system modeling. While the impacts of changing carbonate system properties on biology is often simulated, the feedback effects of that biology on the oceanic and atmospheric carbon are much less often addressed, and may become increasingly important as research into novel fields like carbon dioxide removal increases. Overall, this was a clearly-written paper. While the results are somewhat preliminary, covering the initial setup and some short simulations to verify its ability to replicate current conditions, I think that even as a proof of concept this will be a valuable contribution to the field. There are a few places where the model details and the thought processes leading to those details could be expanded and clarified, which I include in my specific comments below.

We thank the reviewer for their comments on this paper and respond point by point below.

My one broader comment for this paper is in regard to how iCORAL is alternatively framed as either 1) a coral module specific to the iLOVECLIM framework vs 2) a module more generically available for use in ESMs as a whole. Several of the model implementation choices (e.g. subgridding the bathymetry to compensate for low horizontal resolution, adding temperature variability in the tropics) are clearly tailored to iLOVECLIM. Also, the source code is not written in a standalone, modular manner (it loads many of its variables through external fortran modules from the wider iLOVECLIM code*, many parameters are hard-coded, etc.); porting to a new ESM would be a non-trivial task. I suggest revising the text where relevant to clarify that the algorithms and concepts behind the model may be applicable more widely, but significant work would be necessary to port, test, and validate the model under a new ESM coupling.

We have modified the text to make it clearer that this is a module specifically adapted to the iLOVECLIM model, but which is largely reusable in other ESMs.

*Regarding the source code, the author comments (CEC1) imply that they made iLOVECLIM source code available to referees, but I did not receive any documents apart from those available for public review. Therefore, I have not been able to check whether the coral_mod_paper.f90 code (as linked in the Code availability section) compiles and runs in

that context. However, the coral_mod module itself is clearly written -- well-commented, cleanly organized and formatted -- and seems mathematically sound.

We have made a repository with the iLOVECLIM code available to the reviewer. The link to this repository has been given to the topical editor who should be able to provide it. As discussed in CEC1 we hope to be able to have the entire model code publicly available in the future but this is not the case at present due to unresolved licensing issues for several parts of the code.

In addition, we have created an offline version of the iCORAL module which is avalaible on zenodo (doi: 10.5281/zenodo.10932293).

**

SPECIFIC COMMENTS

**

Section 2.1: The iLOVECLIM setup description could use a bit more detail. First, I wasn't entirely clear what the relationship between iLOVECLIM and LOVECLIM was. Based on the iLOVECLIM GMD special issue text (https://gmd.copernicus.org/articles/special_issue30.html), I gather that iLOVECLIM derived from LOVECLIM, minus ice (and possibly minus ocean carbon/bio?), then developed independently? A quick summary of this history in the paper would be useful to orient readers. Also, is the ocean carbon cycle model described in this paper a standard part of the iLOVECLIM configuration? I believe you're using the HAMOCC model for ocean biogeochemistry; can you specify which version? Are there any other parts of your iLOVECLIM configuration (other than the addition of iCORAL) that differs from other published implementations?

As noted by the reviewer, the only difference between iLOVECLIM and LOVECLIM in the version used is the carbon cycle module. The ice sheet is indeed different but it is not used in this version. We have made this clearer in the text:

"The ice sheet module (not used in this version) and the ocean carbon cycle module (used in this version) differ from LOVECLIM (Bouttes et al., 2015)."

The rest of the model, apart from the new coral module, is standard. The carbon cycle module used is the standard version of iLOVECLIM coming from HAMOCC3.1. We have made this clearer:

"The ocean carbon cycle, which is the standard carbon cycle module of iLOVECLIM and described in Bouttes et al. (2015), is based on a Nutrient-Phytoplankton-Zooplankton-Detritus (NPZD) model (HAMOCC3.1, Six and Maier-Reimer, 1996; Brovkin et al., 2002)."

Line 91-92: "Photosynthesis takes place in the euphotic zone in the upper 100 meters." Can you clarify whether this depth is prescribed or an emergent property of the model?

It is prescribed: "Photosynthesis is prescribed in the euphotic zone, set as the upper 100 meters."

Line 95: Given that iCORAL is an extension of ReefHab, a quick description of that model would be useful. It sounds like ReefHab is a classic coral production model, and iCORAL uses the same basic equations but adapted to use ESM-derived input rather than just observational datasets (and with some expanded constraints like the bleaching addition.) Saying this explicitly would help orient readers unfamiliar with ReefHab.

Our model description includes the description of ReefHab, so that the reader does not have to be familiar with ReebHab. We have specified the parts of the code that are derived from ReefHab and those that are new developments and have added more information as follows:

"As in ReefHab, iCORAL first computes…"

"The nutrient and salinity thresholds utilised in the coral module are similar to those of ReefHab. The thermal limits however use the temperature in each grid cell at each depth unlike ReefHab which only uses sea surface temperatures."

"This equation expands on that used in ReefHab, which was similar but without $f_T(T)$, $f_O(\Omega)$ and $f_B(bleach)$."

Figure 1. I didn't find this figure particularly useful; the graphical elements don't provide any new information. I suggest either turning this into a more information-dense schematic (perhaps diagraming the input from coupled model components vs. offline datasets, and the additional pre-preprocessing steps applied before feeding that data into iCORAL, etc.) or eliminating it.

As suggested we have removed this figure.

Line 110: Perhaps clarify that the production depth is defined as the depth at which light is at the Imin level? Readers unfamiliar with the typical light attenuation equation may not immediately parse this detail.

As suggested we have added this clarification:

"The production depth is defined as the depth at which light is at the $I_{min}$ level."

Line 115: Please provide more detail about how the K490 coefficients were chosen and calculated. What specific MODIS data was used to construct the binned data? Did you use the entire mission composite, and if so, through what dates? At what horizontal resolution (4km or 9km)? How were these data matched to the iLOVECLIM horizontal grid? Other satellite-based algorithms are available to calculate this attenuation coefficient near turbid water (for example, NOAA's version based on Tomlinson et al., 2019: https://doi.org/10.1080/2150704X.2018.1536301); were these considered?

We have used the entire mission composite at 9 km resolution, encompassing 15 years from 2002 to 2016. This has been regridded on the CLIO grid (3° by 3°). We have not considered other satellite-based algorithms. Although it would be interesting to study the differences obtained using other products, with the resolution of the iLOVECLIM model we expect this would be second order. Were the coral reef module implemented in a higher resolution ocean model, it would likely be interesting to evaluate the effect of different satellite-based algorithms.

We have added the information about the MODIS data in the text:

"The MODIS data are taken from the entire mission composite at 9km resolution, encompassing 15 years from 2002 to 2016, and has been regridded on the CLIO grid (3° by 3°)."

Also, I'm curious about the decision to use a satellite product to derive these coefficients, rather than deriving the attenuation directly from the model itself (i.e. as a function of chlorophyll + clear-water attenuation, with the latter tied to the simulated phytoplankton group). Given the intent to use this model for paleo-scale simulations, I'm concerned that this choice may decouple the model's simulated chl from the prescribed attenuation coefficients that are at least in part a function of satellite-era chl concentration. I recognize that using a prognostic base for this calculation comes with its own set of difficulties (requiring skillful phytoplankton simulation and some estimate of the CDOM/sediment/etc. distribution), but I think this choice warrants some additional discussion in the paper. You do later mention this as a caveat in the Discussion section 4.1 (Model caveats), but I'd like to see it addressed more fully, either in the Methods or in that caveats section.

We agree it would be better to compute it in the model, but with the low resolution of iLOVECLIM and the use of a simple NPZD, it is likely that the computation will bring biases and probably not improve the module overall for the present day. We however think that this should be tested, in particular in a higher resolution model. We have added discussion in the "model caveats" section with respect to this:

"As iLOVECLIM has low resolution and includes a simple NPZD model, computing the attenuation would likely add biases to the model results for the present-day climate. It should nonetheless be tested in future studies, and in particular if the module was included in a higher resolution ocean model and for use in different climates and land configurations. This will be tested in future work."

In addition, following the second reviewer's advice (Andy Ridgwell), we have tested the use of a homogeneous $K_{490}$, further details are given in the response to this comment.

On a related note, is the iLOVECLIM configuration capable of responding to long-term changes in wetting/drying? I.e. if sea level rises, can grid cells previously designated as land become inundated, or vice versa? If so, how does the iCORAL model handle parameterization of grid cells without satellite-era data to use? Given that an inundation-related scenario is presented as a driving theory for the potential importance of coral reefs in the carbon cycle, this is a key technical point in the potential usefulness of this model setup.

We have recently changed the iLOVECLIM code to modify the bathymetry during a simulation (Bouttes et al., Climate of the Past, 2023). When a grid cell previously considered as land becomes ocean, its ocean variables are initialised with the closest neighbours' values. This will be adapted for the coral reef module in future work, and we will also likely test computing the attenuation coefficient so that it can evolve in very different configurations and climates.

Line 155-160: The description of the 1-m vertical subgridding could be clearer; it took me a few readings to figure out what was being done. Using terminology like "vertical resolution"

and "depth interval" was a little misleading to me, since initially I interpreted this as a sub-gridding of the water column, rather than a pairing of each 3-deg grid cell with its subgrid-scale bathymetry distribution.

We have added more details to make this clearer:

"Because the vertical resolution in the model is relatively coarse (increasing from 10 meters at the surface to 28 meters at 100 m depth), coral production is computed on a sublevel vertical grid every meter. This allows us to account for the fine vertical changes in light attenuation, surface availability and bathymetry. The other variables, taken from the ocean model, are homogenous in an ocean grid cell (temperature and aragonite saturation state). The carbonate production at 1-meter vertical resolution is then aggregated in each ocean cell."

In addition, following the comments from Reviewer2 we have added an additional figure with the vertical grids.

Line 243: Criv and Ariv are never explicitly defined; I assume those are the river input flux of DIC and Alkalinity, respectively?

Yes, we have added this in the text.

Line 259: "AR(p)" may be too jargon-y for a non-statistical-modeling audience; please provide a quick definition of this model type and what the model order (p) refers to. Also, you mention supplementary material here, but I could not locate any supplementary material associated with this manuscript.

We inadvertently forgot to include the relevant memo in the supplementary material when preparing the submission. That document (now uploaded in comment AC4: https://doi.org/10.5194/egusphere-2023-1162-AC4) will be included with the revised manuscript's Supplement. It provides references to the exact dataset used and also provides a detailed description of the processing, as well as the results for each of the six AR models fitted. We nevertheless provide a (very) short outline about the essence of AR(p) models. The text was rewritten as follows:

"We have thus generated additional temperature variability, based upon the analysis of the daily sea surface temperature anomalies in a tropical region with extended coral reef cover (19–16°S, 148–154°E). We fitted a series of autoregressive models of order $p$, denoted AR($p$) models ($p$ = 1, …, 6) to the daily time series in each grid point in this area. An AR($p$) model predicts the value of a variable at time $t$ as a linear combination of the $p$ previous values plus random noise. The fitting procedure provides the parameter constants for the linear combination (i.e., autocorrelation parameters – for details about the dataset used and the processing steps, please see the *"Autoregressive Model to Parametrise Temperature Variability"* memo in the Supplement). Here, we selected the AR(1) model, as the RMSEs of the higher order models were not statistically different. Accordingly, we generate an AR(1) variate with an auto-correlation parameter of 0.90 and a Gaussian distributed random noise with a standard deviation of 0.28 to add daily variability to the otherwise anomalously smooth temperature evolution in iLOVECLIM."

Line 263: Where did the values of 0.90 and 0.28 come from? Are they standard values or were they derived from the fitting process?

These values were derived during the fitting process. This is now more clearly stated in text (see reply to previous comment).

Figure 4: Anomaly plots would be helpful to allow quicker direct comparison between column 1 and 2.

As suggested we have added a column with the anomaly to facilitate the comparison between data and model.

[Figure]

**Figure 3. Model (left), observational data (middle) and model-data difference (right) surface maps of (a, b, c) temperature (°C), (d, e, f) salinity, (g, h, i) phosphate (μmol kg⁻¹) and (j, k, l) aragonite saturation state (Ω). The model outputs are averaged over 2000-2010. The data are from Locarnini et al. (2018), Zweng et al. (2018), Garcia et al. (2018) and Jiang (2015). The model outputs have been regridded on the data grid to compute the anomaly.**

Lines 319-322: You mention that coral overproduction in the model might be due to lack of riverine nutrient input leading to lack of competition from macroalgae. While that is one possible source, lower-resolution models (even 1-deg) tend to underestimate coastal pelagic production due to poor resolution of shelf dynamics, which could alter the conditions in these regions. And it doesn't appear that this model simulates macroalgae at all, so even with improved riverine input that competitive pressure would be missing in the simulations. A bit more discussion of the possible drivers of oversimulation of coral could be useful here.

This model does not simulate macroalgae, but the indirect effect of abiotic conditions on the competitive advantage of macroalgae is partially accounted for by limiting coral habitability to grid cells where the phosphate concentration is below 0.2 μmol L⁻¹, as done previously in ReefHab.

Line 323: "might not be present in the observed data."  Are you saying that these types of features might exist in the real world but not be captured by this particular dataset?  Or that the model is simulating a type of isolated coral reef that probably doesn't exist in the real world?

Indeed, we are saying that such isolated small coral reefs might not be captured in observational products. But it is equally possible that other limiting factors may prevent coral reef development in these regions and that the model is over-predicting coral reefs there. We have made this clearer in the text:

"In addition, the model also simulated small isolated coral reefs with small areas (in purple) that might not be captured in the observed data. Alternatively, other limiting factors, not represented in iCORAL, might prevent coral reefs to develop in such areas."

Lines 432-434: This statement implies that perhaps the answer to my inundation question above is no, the model cannot handle wetting/drying.  It's not clear to me how one would reparameterize coral growth parameters to be dependent on sea level without explicitly allowing for this; can you expand on this idea?

As previously discussed, we have modified the code to allow for bathymetry changes. However, this is constrained to periods when we have bathymetry reconstructions. As we intend to use this model for periods when such reconstructions do not exist, we also plan to test changing the bathymetry by simply modifying the sea level. We will then compare the results using both methods for the periods with bathymetry reconstructions (Last Deglaciation) to evaluate differences between the methods. This will be performed in future work beyond the scope of this paper, and is also discussed as a response to reviewer 2.

**

TECHNICAL COMMENTS

**

Equations: Equation numbers would be nice (possibly not a requirement for this journal?).

We have added equation numbers.

Line 266, 277, etc: Use of present perfect rather than past tense read a little strangely to me.

We have changed the text here to the past tense.

Figure 2: Is this just presence/absence data?  Please clarify in the caption or add an appropriate legend.

Yes, it is just the presence of coral reefs, we have modified the caption:

"Coral location from UNEP-WCMC (2018) dataset. Brown cells indicate the presence of coral reefs in these cells. In white grid cells no coral reef has been detected."

Fig. 7a: I suggest adding the blue/purple highlights to this panel for consistency with Fig 7b, 7c, 8, and 9.

This has been added.

---

## Author Comment (AC6)

**Reply to RC2**

Review of:
Implementing a coral reef CaCO3 production module in the iLOVECLIM
climate model

by Nathaelle Bouttes and co-authors / egusphere-2023-1162

Coral reefs and associated controls on carbonate precipitation and burial and a potentially key, but to date overlooked, interactive carbon cycle component in Earth system models. To my knowledge the authors are right in that previously, only prescribed, rather than interactive, carbon and alkalinity sinks associated with shallow carbonate production have been implemented in (3D ocean-based) global models. As such, the current work represents a very useful modeling advance and highly appropriate for the timescale capability of the 'iLOVECLIM' climate (Earth system?) model. The paper is well-written and the model parameterizations generally well described and justified. I do have a number of minor comments (listed later). However, I do also have some questions about whether some of the assumptions made in the construction of the coral reef CaCO3 production module tie carbon and alkalinity feedback too closely to the modern marine environment and observations, preventing direct past (geological) applicability and potentially also somewhat limiting future capabilities of the new coupled model.

We thank Andy Ridgwell for his comments and suggestions. We have replied point by point below in blue.

**Generalizability/applicability of the model**

A couple of assumptions are made in the coral reef CaCO3 production module have implications for its applicability to non-modern, and particularly paleo situations.

1. Diffusive attenuation coefficient

As an initial note – I think URLs are not allowed these days(?) I did go to the page and try and retrieve the data, but either I was being incompetent, or the details given in the text are insufficient to retrieve the specific data in question. Ideally, the retrieved data would be placed somewhere with a DOI. I did check the DOI given for iCORAL, but it is only the FORTRAN file and does not include any boundary conditions. A DOI is in any case needed for the current version of iLOVECLIM, and that could then include relevant boundary conditions such as for the K490 field(?)

We have added more information on the used MODIS product in the manuscript and have deposited the regridded file on zenodo where it is available with a doi:  10.5281/zenodo.10776565:
"$K_{490}$ is the diffuse attenuation coefficient at 490 nm taken from the Level-3 binned MODIS-Aqua products in the OceanColor database (available at: http://oceancolor.gsfc.nasa.gov). The MODIS data are taken from the entire mission composite at 9km resolution, encompassing 15 years from 2002 to 2016, and have been regridded on the CLIO grid (3° by 3°).

So my question is: how important is the diffusive attenuation coefficient field? If a mean global value was applied uniformly, or representative open ocean value applied uniformly, how different does the projected distribution of reefs and global carbonate production become?

Using the present-day satellite-retried spatial pattern potentially strongly pins the modelled distribution to 'now'. In the future with changing river flow, sediment loads, etc., the pattern may change, introducing a bias in future simulations. Much more problematic would be paleo applications, particularly when the land-sea mask is different and one can no long map present-day satellite retrievals onto past oceans. What are the author's plans for applying iCORAL-iLOVECLIM to the geological past and what are they planning to assume re. K490?

My guess, given that the baseline model (Figure 5a) struggles with e.g. correctly projecting the absence of reefs in the NW Atlantic anyway, is that high sediment loads and the absence of hard substrates may be more important than getting light attenuation 'right'. Hence I wonder whether one could apply a mean or representative value globally, accept a small degradation in model fidelity, but remove this tie to the present-day?

This is indeed important for future applications in climates that strongly differ. However, using a homogeneous constant $K_{490}$ value, or a geographical 2D $K_{490}$ variable that is constant in time should have limited impact in different climate states. The main improvement would be to compute $K_{490}$ using nutrient and productivity, but this is beyond the scope of this study. We plan to try this in the future, in particular with the coral reef module implemented in a higher resolved model.

Nevertheless, we have still tested using a homogenous $K_{490}$ value. We have chosen the mean $K_{490}$ value between 30°S and 30°N: $K_{490} = 0.041$. We can compare this simulation with fixed homogenous $K_{490}$ with the 'best case' simulations presented in the paper.

The simulated coral distribution is similar in both cases (see figure below) but the production value is increased with the fixed $K_{490}$, resulting in higher global production (1.02 Pg $CaCO_3$/year) compared to the 'best case' simulation (0.82 Pg $CaCO_3$/year).

While using a scalar or a 2D matrix for $K_{490}$ will not improve simulations for different climates, using a 2D field improves comparisons with local modern data. We thus prefer to keep a 2D $K_{490}$ field as it permits a better representation of observations and allows us to better identify alternative reasons for possible mismatches.

| | Area (1e3 km3) | CaCO3 production (Pg CaCO3/year) |
|---|---|---|
| Reference simulation (best case) | 390 | 0.82 |
| With fixed $K_{490}$ | 490 | 1.02 |

[Figure]

| Coral location in the 'best case' simulation | |
|---|---|
| Coral location with fixed $K_{490}$ | |
| | model-data agreement on coral presence |
| | coral from data not simulated by model |
| | coral simulated by model not observed in data |

| | |
|---|---|
| Production anomaly (g/year) between the Kd fixed simulations and the 'best case' simulation |
[Figure]
 |

2. Sea-floor bathymetry

I understand exactly why the authors have imposed a much higher resolution sea-floor bathymetry on the reef module. However, while for far future simulations one could simply take into account a mean sea-level change, things become (isostatically) more complicated if you go back to e.g. the last glacial maximum, and I am sure that (and the glacial-interglacial cycles in general) will be a scientific target of the authors.

If the paleo questions were restricted to the last glacial cycle, then relatively high resolution (10 minute) reconstructions are available, e.g. ICE-6GC, GLAC-1D, as per PMIP4. What would the coral reef coverage and carbonate production look like if the 0 ka dataset from ICE-6G-C was used? If the authors plan deglacial (ICE-6G-C) or penultimate deglaciation (GLAC-1D) applications, if would be worth-while in the current paper calibrating and evaluating a slightly lower-resolution pale-enabled version of iCORAL/iLOVECLIME using e.g. ICE6G-C bathymetry data.

Moreover, I cannot help but wonder what the results of simply using the iLOVECLIM ocean grid would be. Sure, reef locations would be very patchy, but as long as there was some sort of distribution of reef occurrence between Pacific, Indian, and Atlantic Ocean basins, I see no reason why the feedback between climate and carbonate removal should not be equally plausible (given a tuning resulting in a plausible initial global carbonate burial flux). This would make iCORAL-iLOVECLIM generically (and equally) applicable past global carbon cycle/climate questions.

As an aside – I did not see the bathymetric resolution stated. The authors state that they bathymetry comes from '*GEBCO 2014*' and cite '*GEBCO Compilation Group, 2022*'. Going to the GEBCO Compilation Group website, the current data-set is 2023 and at a resolution of 15 arc-seconds. No dataset further back than 2019 is available that I could see and so I am unsure what '*GEBCO 2014*' refers to. So a little more detail on the dataset used is needed.

We used the GEBCO_2014 data released in 2015 and available here: https://www.gebco.net/data_and_products/historical_data_sets/. We have added the website reference in the manuscript to make it easier to find.

In addition, we have tested using the ICE-6G_C bathymetry (we expect the results would be similar with the other LGM reconstruction GLAC-1D) instead of GEBCO to compute the subgrid available area at each vertical level (see figures below). Using the ICE-6G_C bathymetry results in a larger area covered by coral reefs ($908 \times 10^3$ km$^2$ compared to $390 \times 10^3$ km$^2$ with the GEBCO bathymetry). The global production is also higher with 2.26 Pg CaCO$_3$/year. Tuning the model to obtain results within the range of observed data should be doable. However, the location of coral reefs is degraded, with more grid cells where the model does not simulate coral reefs contrary to observations. Because the bathymetry resolution is degraded, we miss areas where coral reefs should develop.

Hence with ICE-6G_C the coral location is still first order accurate, but there are more mismatches with the observed data. As discussed by the reviewer, our anticipated work will indeed focus on the future Anthropocene and glacial interglacial changes. We intend to first use a passive submersion version, where only the eustatic sea level change is accounted for. We also envision testing the ICE-6G_C bathymetry, but as the reviewer points out, this only covers the last 26 kyr. We plan to use both methods for the last deglaciation (passive submersion with the GEBCO bathymetry and evolving ICE-6G_C bathymetry) and compare approaches.

We do not plan to consider deeper time periods. For such deep past periods the bathymetry changes more and it might be better to use the reconstructions, despite their low resolution. As we will confine our studies to glacial-interglacial periods, we think it is better to prioritize resolved bathymetry with passive submersion (and the ICE-6G_C or GLAC-1D for the last deglaciation).

| | Area (1e3 km$^3$) | CaCO$_3$ production (Pg CaCO$_3$/yr) |
|---|---|---|
| Reference simulation ('best case') | 390 | 0.82 |
| With ICE-6G_C bathymetry | 908 | 2.26 |

| | |
|---|---|
| Coral location in the 'best case' simulation |
[Figure]
 |
| Coral location with ICE-6G_C bathymetry | |
| | model-data agreement on coral presence

coral from data not simulated by model

coral simulated by model not observed in data |
| Production anomaly (g/year) between the simulations | |

3. Temperature variability in iLOVECLIM

The 3rd assumption that ties iCORAL to the present-day is the imposition of enhanced sea-surface temperature variance in the tropics. Again, I can see the reasoning behind this, but some details are missing. In particular, the text says: '*for details, please see supplementary information*' but I could not find the SI anywhere.

Infeed, we forgot to add the SI. As also replied to reviewer 1 we have added a specific comment for this, see comment AC4 (https://doi.org/10.5194/egusphere-2023-1162-AC4).

How big an effect is this? Is it a relatively small effect, or is it fundamental to getting the distribution of reefs and global carbonate sink anywhere near correct (a comparison would be helpful to see and I suspect informative to readers)? If the former – could not the bias imposed by adopting un-adjusted iLOVECLIM climatology be 'tuned away' ? If the latter – what confidence do the authors have in future and past applications? I was under the impression that variance may change in the future. If only a little, then this may not matter. But what about the last glacial, or the Eemian? Would SST variance be expected to be more, less, or about the same? i.e. how safe is the assumption of observationally-derived SST variance in the past?

Lastly, why only restrict the modification to the tropics? Why not globally? I guess one answer is that there are very few reefs outside of ±30 ∘ (Figure 2). However, rather more model-projected reef locations occur outside of the tropics (Figure 5a,b), and there will potentially be a very different (and spurious) bleaching response either side of the boundary.

I think in general and across all this points above – firstly, knowing the importance (or not) of making the various assumptions and imposing boundary conditions derived from modern observations would be informative and helpful. Secondly, the more that iCORAL can utilized internal iLOVECLIM fields and boundary conditions, the more generally applicable it will be to the future and particularly the geological past. If the authors do not want to make the choice between more 'realistic' and modern-tied vs. a poorer fidelity simulations of present-day reef distributions and global carbonate productions, then why not calibrated, evaluate, and present, two (or more) alternative setups and calibrations that could be used with iLOVECLIM applied to different questions? Overall, many of the choices and assumptions made in developing iCORAL seem to be orientated towards reproducing observations rather than enabling carbon-climate feedback and the stated aim of 'past and future coral-climate coupling'.

We realize that this was not correctly formulated in the manuscript: on line 257, "in the tropics" is incorrect and must be deleted. Temperature variability was actually added everywhere, but the AR model *parameters* were calibrated on data from tropical region with extended coral reef cover (as correctly stated on lines 258-259. This paragraph was partly rewritten, also in response to a comment by Referee #1 (see reply to their comments)

For the pre-industrial steady-state distribution, temperature variability affects neither location nor productivity, as bleaching is never triggered. This will play a role when temperature will increase, as is the case in future simulations following the SSP scenarios. This will be the subject of future work and is beyond the point of this paper.

**Model fields and coral reef location evaluation**

I think missing is a sufficiently critical discussion of the model fields driving iCORAL (Figure 4). To me, the surface ocean saturation is rather lower in the tropics in iLOVECLIM vs. observations, while nutrients – which are assumed to prevent reef formation above a threshold – are higher. (Note that the depth of the 'surface' layer is not given in the figure and needs to be.)
As suggested, we have added the depth of the model surface layer in the figures (5 m).

There are more localized mismatches in temperature and salinity which may or may not also play a role. I am not at all concerned about the existence of model-data mismatches, which is par for the course, but rather that their potential implications are not sufficiently discussed. 3 parameters are tuned and I wonder to what degree they are countering errors in the simulated environmental fields. In all biogeochemical modeling of this sort, the risk is always that you correct for a deficiency by distorting something else, with the potential that e.g. the strength of carbonate-climate feedback could end up very different.

I think that at the very least, more discussion about how biases in certain simulated environmental parameters and regions might impact projected reef locations. Further evaluating iCORAL by feeding it observed fields (Figure 4a) in place of simulated fields (Figure 4b) would be interesting. Replacing fields one-by-one might be further instructive. One could do this comprehensively, potentially even retuning iCORAL for each combination of simulated or observed environmental fields. Or it might be sufficient in the paper simply to take iCORAL as it is (and its current tuning), and test swapping out the simulated for observed fields.

In addition, there may be better ways of comparing simulated and observed fields (Figure 5). For instance, for each observed reef location, one could pull out the simulated and observed values at those locations and cross-plot, perhaps color-coding for basin. Or color-code as per in Figure 6 and pulling out both 'real' and simulated locations. This would be a way to try and identify whether there are any specific model environmental biases which tended to generate false positives or negatives in reef location.

The more you can pull out specifically why – in terms of simulated environmental conditions or model parameters or structure – false positives or negatives occur, the more we'll learn.

The results of the coral module will indeed depend on the environmental fields simulated by the climate model. As in every coupling, the parameter tuning is strongly dependent on the input variables, here temperature, salinity, phosphates, light, aragonite saturation state. As suggested we have added more details on the model biases and their possible impact on coral reef development:

"While the coral reef model could be best calibrated and compared to observations using present-day environmental conditions, we aim for iLOVECLIM applications to climates far beyond the current state. Therefore, we use a dual approach. We test the model using best observational drivers but make sure that we could link these drivers to internal model variables or use simplified approaches applicable for wide range of climates. However, the coupled model application to other climates is beyond the scope of this paper."

"The sea surface temperature in the model is generally slightly higher than in the observations, especially in the tropics where it can be 2°C higher than in the observations. The coral reef development is limited by a maximum temperature, which could be reached quicker than in observations due to the high temperature bias. The distribution of simulated nutrients exhibits greater biases. The concentrations simulated by the model are generally low compared to observations, especially in eastern equatorial upwelling regions. The resulting effect is the opposite as the one due to the temperature bias: the coral reef development will be less affected by phosphate changes as the maximum limit is further away due to the lower phosphate bias. The saturation state is also in generally good agreement with data, despite some differences locally. In particular it is slightly higher than the observed values in the tropics."

The tuning will necessarily compensate for any model bias. It also depends on the model resolution. To give an idea of how important this is we use an offline version of the coral module forced by observed annual fields of the input variables (as presented on Figure3). We have added the offline module in the data section of the manuscript with a doi: 10.5281/zenodo.10932293.

We have run a simulation of coral production with a 1degree resolution grid, and a set of simulations with a 3degree resolution grid. The horizontal iLOVECLIM resolution is also 3degrees. With the 3 degree grid we first use observed

fields for the inputs, then use the SSTs from iLOVECLIM only, then the PO4 from iLOVECLIM only (the other inputs fields are still from observed data).

The simulated surface area is of the same magnitude with iLOVECLIM and the 3-degree grid offline model. The production is halved when using the offline version. Changing from 3-degrees to 1-degree horizontal resolution results in a decrease of both surface area total production. When swapping the observed temperature field by the iLOVECLIM temperature, the area and production are decreased, as the iLOVECLIM SSTs are higher, hence more limiting. It is the opposite when swapping the PO4 field for the iLOVECLIM one, as the PO4 in iLOVECLIM is lower than in the observations.

| | iCORAL in iLOVECLIM | iCORAL offline | | | |
|---|---|---|---|---|---|
| | | With 3deg observed fields | With 3deg observed fields + SST from iLOVECLIM | With 3deg observed fields + PO4 from iLOVECLIM | With 1deg observed fields |
| Area (1e3 km$^2$) | 390 | 420 | 240 | 540 | 300 |
| annual CaCO3 production (PgCaCO3/year) | 0.82 | 0.40 | 0.21 | 0.51 | 0.28 |

[Figure]

| iCORAL in iLOVECLIM | |
|---|---|
| iCORAL offline, 3 degrees with observed fields | |

[Figure]

| | |
|---|---|
| iCORAL offline, 3 degrees with SST from iLOVECLIM | |
| iCORAL offline, 3 degrees with PO4 from iLOVECLIM | |
| iCORAL offline, 1 degree with observed fields | |
| | model-data agreement on coral presence |
| | coral from data not simulated by model |
| | coral simulated by model not observed in data |

**Minor comments[1]**
* * *
1 Suggested text changes indicated with → and suggested inserted words underlined. **x** represents line number.

{Suggested text changes indicated with → and suggested inserted words underlined. x represents line number.}

- **18** – 'feedback' would be a better (much more common) word than 'retroaction'.
  Changed as recommended

- **24-25** – '*The model enables assessment of past and future coral-climate coupling on seasonal to millennial timescales*' – just noting the aim in the context of the present-day assumptions and my comments above.
  We have added a reference to the model limitations:
  "The tuned model simulates the presence of coral reefs and regional-to-global carbonate production values in good agreement with data-based estimates, despite some limitations due to the imperfect simulation of climatic and biogeochemical fields driving the simulation of coral reef development. The model enables assessment of past and future coral-climate coupling on seasonal to millennial timescales, highlighting how climatic trends and variability may affect reef development and the resulting climate-carbon feedback."

- **52-53** – You don't have to add them, but just pointing out some empirical / machine learning papers: Couce et al., Future habitat suitability for coral reef ecosystems under global warming and ocean acidification, Global Change Biology DOI: 10.1111/gcb.12335 (2013); Couce et al., Tropical coral reef habitat in a geoengineered, high-CO2 world, GRL 40, doi:10.1002/grl.50340 (2013).
  This has been added as suggested.

- **87-88** – Without digging out Millero (1995), I can't remember whether it included anything about solving the carbonate system or not. If not, missing are details of the numerics. Millero (1995) is also full of typos in various dissociation constant coefficients, so there must be a better reference for what the authors have adopted in terms of e.g. dissociation constants.
  The reference to Millero (1995) was removed (it does indeed not include anything about solving the carbonate system) and the text modified as follows:
  "Surface ocean $p\mathrm{CO_2}$ is computed from temperature, salinity, DIC and ALK using the polynomial ACBW solver from SolveSAPHE (Munhoven, 2013), updated to revision 1.0.3 (Munhoven, 2020), with the $\mathrm{pH_{SWS}}$ configuration."

- **89** – '*nitrates*'? Do you mean: nitrate and ammonia. Or nitrate and ammonia and dissolved N2? Or just NO3, which would be singular 'nitrate'?
  This was a typo and has been corrected ('s' removed).

- **107-109** – Please add a brief justification for the limits. e.g. I think the northern Red Sea reaches 41 PSU around the Gulf of Suez (Google further tells me that there are 35 coral taxa in the Gulf of Suez). For phosphate – is this a real-world threshold, or chosen in light of the iLOVECLIM surface nutrient simulation? Looking at the WOA annual mean surface PO4, locations incorrectly simulated in the model in Figure 5a lie in surface waters with PO4 above 0.2 – here I am looking at the NW Atlantic and SE Pacific. Is plankton productivity (and turbidity) not the more proximal factor influencing the presence/absence of corals (with nutrient availability influencing productivity)?
  I noticed that only later down the text does it state: '*The nutrient and salinity thresholds utilized in the coral module are similar to those of ReefHab.*' It would still be helpful to know a little more on the justification, and how important these assumptions are in leading to e.g. Figure 5a.
  The limits are the same as in ReefHab, which allows us to compare our new results with their previous results. To have more information on the justification of the limits we refer to Kleypas et al. (1999) and references therein, which we have added in the manuscript:
  "The habitability is based on modern observations of coral presence and environmental conditions (Kleypas et al., 1999b and reference therein)."
  Kleypas, J. A., McManus, J. W.C. and Menez, L. A. B.: Environmental Limits to Coral Reef Development: Where Do We Draw the Line?, Am. Zool. 39 (1), 146-159, doi:10.1093/icb/39.1.46, 1999.
  In the future, the coral reef module could be improved by changing the habitability limits, but this is beyond the scope of this paper.

- **Section 2.2.2** – A schematic of the gridding and grid relationships would be helpful. Maybe pick a single illustrative region and show of he grid relate, both horizontally and vertically. This could be combined with Figure 1 as a second panel (or thrown in SI).

  As suggested we have created a new figure to detail the vertical model grid cell and the subgridding. This has been added as supplementary figure 1.

[Figure]

Figure S1. Schematic of the model vertical grid (m) with centre in blue dots and interface in red lines, and the 1m vertical subgrid (black lines) used for computing coral reef development.

- **150** – $K_{\text{arag}}$ could be confused with $K_{\text{sp}}$ (of aragonite) to a sloppy reader like myself. If it is a saturation value (reference value or threshold), why not $\Omega_{\text{ref}}$ or something?
  $K_{\text{arag}}$ has been replaced by $K_{\text{omega}}$ in the revised manuscript to avoid the kind of confusion mentioned.
- **164-165** – Text describing the relationship between grids, gradients, etc. would be much clearer with a figure (see earlier comment).
  As suggested we have created a new figure (see response to related comments earlier)
- **196** – It is a shame there there is not a DOI or anything less nebulous than a webpage ('*https://www.coralreefwatch.noaa.gov/product/5km/methodology.php*').
- **Section 2.2.4** – I don't know why this doesn't come across clearer. It is correct (in terms of DIC and ALK relationships and flux balance), but a little round-about.
- **Section 2.2.5** – See comment on present-day observationally-tied temperature variance.
- **269-270** – Maybe make this clearer earlier in the text (see earlier comment).
  This has been amended in response to the earlier comment (see above).
- **452** Given iCORAL is embedded within iLOVECLIM, we need a DOI for the version of iLOVECLIM used (indeed, the code for iCORAL utilizes a number of iLOVECLIM modules and the iCORAL code is insufficient in isolation).
  We do not have the rights to publish a version of iLOVECLIM, but in addition to the added coral module files that are available on zenodo, we also provide an offline version, also on zenodo (doi: 10.5281/zenodo.10932293) , of the coral reef module. This has been added in the manuscript.